# Chinese *Wu*, Ritualists and Shamans: An Ethnological Analysis

**Michael James Winkelman** 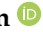

School of Human Evolution and Social Change, Arizona State University, Tempe, AZ 85281, USA;
michaeljwinkelman@gmail.com

**Abstract:** The relationship of *wu* (巫) to shamanism is problematic, with virtually all mentions of historical and contemporary Chinese *wu* ritualists translated into English as shaman. Ethnological research is presented to illustrate cross-cultural patterns of shamans and other ritualists, providing an etic framework for empirical assessments of resemblances of Chinese ritualists to shamans. This etic framework is further validated with assessments of the relationship of the features with biogenetic bases of ritual, altered states of consciousness, innate intelligences and endogenous healing processes. Key characteristics of the various types of *wu* and other Chinese ritualists are reviewed and compared with ethnological models of the patterns of ritualists found cross-culturally to illustrate their similarities and contrasts. These comparisons illustrate the resemblances of pre-historic and commoner *wu* to shamans but additionally illustrate the resemblances of most types of *wu* to other ritualist types, not shamans. Across Chinese history, *wu* underwent transformative changes into different types of ritualists, including priests, healers, mediums and sorcerers/witches. A review of contemporary reports on alleged shamans in China also illustrates that only some correspond to the characteristics of shamans found in cross-cultural research and foraging societies. The similarities of most types of *wu* ritualists to other types of ritualists found cross-culturally illustrate the greater accuracy of translating *wu* as "ritualist" or "religious ritualist."

**Keywords:** *wu*; shaman; ethnological analogy; priests; mediums; healers; witch; China; evolution of Chinese religion; sociocultural evolution of religion

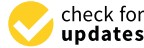



## 1. Introduction: What Is the *Wu*?

The term *wu* (巫) has been widely applied to ritualists of China's past and present (Boileau 2002; Cai 2014; Hopkins 1945; Lin 2009; Michael 2015; Qu 2018; Schafer 1951; Sukhu 2012; Xing and Murray 2018; Fu 2022). The spectrum of *wu* ritualists ranges from presumed archaic practices that persisted as Chinese society transformed from matriarchy to patriarchy, then to tribal chiefs and ancient kings, and eventually a wide range of historical and contemporary ritualists, including mediums and ancestor worship priests. Whatever the original manifestations and meanings of *wu* were, by the Warring States period (fifth to third centuries BCE), *wu* was widely applied to very different forms of ritualists and virtually all subsequent forms of Chinese religious activity (Michael 2015; Williams 2020).

Following Eliade's (1951/1964) seminal book *Shamanism*, the term shaman began to be applied to the translation of *wu* into English (Michael 2015). Even earlier, Hopkins (1945) and Schafer (1951) translated *wu* into English as shaman, but they also used the terms wizard and witch for *wu*. The increasing practice of Chinese scholars translating *wu* as shaman following Eliade's (1951/1964) seminal book was without critical assessments of whether it was appropriate (see von Falkenhausen 1995; Keightley 1998; Boileau 2002; Williams 2020 for critiques). Boileau addresses the consequences of such problematic uses of the term shaman that lack clear references to established features of shamans and used vague definitions that fail to differentiate shamans from virtually any religious ritualist.

This widespread practice of translation of *wu* as shaman is surprising considering Eliade's explicit rejection of such equivalence (Eliade 1964, pp. 450–54). Eliade discussed

vestiges of China's archaic shamanism in male practitioners (*wu xi* 覡 and physicians *wu yi* 巫醫) but rejected the association of shaman with *wu*, whom he characterized as mediums, noting their possession states as an aberrant form and reflective of shamanic decadence. Michael (2015) notes this disavowal by Eliade of any equivalence of shamanism and *wu*: "Although Eliade discussed many examples of ecstatic flight, vestiges of China's archaic shamanism, he did not associate them with the *wu* . . . By *wu*, he specifically referred to 'the exorcists, mediums, and 'possessed' persons . . . [who] represent the aberrant shamanic tradition' (Eliade 1964, p. 450). Shamanism was a heroic venture, and possession was a decisive manifestation of its decadence" (Michael 2015, pp. 677–78). To Eliade, the loss of shamanism was evident in the features of possession characteristic of the *wu*.

Nonetheless, scholars have ignored Eliade's perspective and tended to use shaman in a very loose way to refer to virtually all Chinese ritualists, a practice that undermines the usefulness of the term and points to the need for a different translation of *wu*. Williams (2020, p. 154) points out that by the Han dynasty (206 BC–220 AD), virtually all forms of supernatural practitioners across China were called *wu*: "*wu* . . . comprises almost everybody who has to do with supernatural phenomena." Mair (1990) also noted that the practice of translating *wu* as shaman has problems stemming from a lack of correspondence with the characteristics of the Siberian shamans, beginning with the centrality of the ecstatic flight and the shaman's ritual with the whole community where the healing rituals emphasized soul recovery. "This is in contrast to the *wu* who were closely associated with the courts of various rulers and who were primarily responsible for divination, astrology, prayer, and healing with medicines" (Mair 1990, p. 35). Allan (2015) notes that many of the diverse ritualists called *wu*—also translated as diviners, conjurers and healers—do not even engage in alterations of consciousness, a defining feature of shamanism. Translations of *wu* as spirit medium (von Falkenhausen 1995, pp. 279–80), magician (Mair 1990, p. 35) and diviner (Boileau 2002) illustrate reasonable alternatives to the blanket use of shaman for all types of *wu*.

These descriptions illustrate that from the beginning of Chinese religion, there were several distinctive types of *wu* ritualists. This wide range of *wu* ritualists suggests that the term has a general meaning as [religious] ritualist rather than as shaman in particular and emphasizes the need to differentiate among types of *wu* rather than translating all of them as shaman. The critical question involves the correspondence of any of the various types of *wu* with empirically established cross-cultural features of shamans. Numerous scholars of *wu* have lamented the lack of objective criteria for determining what is a shaman and its features.

This paper presents such objective criteria from ethnological research to illustrate the cross-cultural features of shamans and their differences from other types of religious ritualists. This research provides an objective etic framework for empirical assessments of resemblances of the various types of Chinese *wu* with other religious ritualists, including a cross-cultural foraging shaman. The usefulness of this approach is illustrated below by showing the correspondence of some types of historical *wu* and Chinese shamans with the etic features of shamans, as well as the correspondence of most *wu* types with other types of religious ritualists found cross-culturally. These comparisons are used to show that most ritualists called *wu* or Chinese shamans do not correspond to cross-cultural features of shamans, but rather correspond to other types of religious practitioners.

## 2. Cross-Cultural Methods: A Derived Etic Model of Religious Practitioners

Rather than using theoretical or ideological principles to arbitrarily define the types of religious practitioners, Winkelman's (1986, 1990, 1992, 2010a, 2022) cross-cultural research project used a grounded method of deriving variables from descriptive ethnographic data and performing quantitative analysis of this information to identify cross-culturally valid religious practitioner or ritualist types. This study is based on a subsample of the Standard Cross-Cultural Sample (SCCS) (Murdock and White 2006) which examined 47 societies worldwide (see Winkelman 1992 for details). This subsample provided 115 culturally

recognized types of religious practitioners, professional occupations thought to have a special capacity for interacting with supernatural beings or power. These practitioners were characterized by variables based on ethnographic descriptions of their practices (for data see footnote[1]). This descriptive data represented as variables (nominal, ordinal) was used to generate a matrix of similarities which was entered into cluster analysis to determine the different types of religious ritualists based on their empirically shared features (see Winkelman 1986, 1990, 1992 for details).

These statistical groupings were used to derive an etic model of types of ritualists that are represented in the SCCS/CosSci[2] variables as Shaman[3] (879), Shaman/Healer[4] (880), Healer (881), Medium (882), Sorcerer/Witch (883) and Priest (884). These variables are referred to with initial capital letters in the text to distinguish these etic ritualist types from common concepts represented in the same words. The common and distinctive features of these etic religious practitioner types were determined by frequency analyses of the variables[5] used in determining the types of ritualists; these features are reported in Winkelman (1986, 1990, 1992, 2010a, 2021a, 2022) and presented in Table 1 and others below.

**Table 1.** Characteristics of Religious Practitioner Types (adapted from Winkelman 2022).

| Ritualist Type | Principal Magico-Religious Activity | Selection and Training | Social and Political Power | Professional Characteristics | Motive and Context |
|---|---|---|---|---|---|
| Shaman (Forager Shaman) | Healing and divination. Protection from spirits and malevolent magic. Hunting magic. Cause illness and death. | Dreams, illness, and signs of spirit's request. ASC induction, normally vision quest by individual practitioner alone in wilderness. | High social status. Charismatic leader, communal and war leader. Makes sorcery accusations. Ambiguous moral status. | Predominantly male, female secondary. Part time. No group—individual practice with community. Status recognized by clients. | Acts at client request for client, local community. Community-wide ceremony at night. |
| Shaman/Healer (Agricultural Shaman) | Healing and divination. Protection from spirits and malevolent magic. Hunting magic and agricultural rites. Minor malevolent activity. | Vision quest, dreams, illness and spirit requests. Training by group. Ceremony recognizes status. | Moderate social status. Informal political power. Moderate judiciary decisions. Predominantly moral status. | Predominantly male. Part-time. Collective group practice, ceremonies. Specialized role. | Acts at client request. Performance in client group. |
| Healer | Healing and divination. Agricultural and socioeconomic rites. Propitiation. | Voluntary selection, large payments to trainer. Learn rituals and techniques. Ceremony recognizes status. | High socioeconomic status. Judicial, legislative, and economic power. Denounce sorcerers. Life-cycle rituals. Predominantly moral status. | Predominantly male, female rare. Full-time. Collective training, practice and ceremony. Highly specialized role. | Acts at client request in client group. Treatment in client group. Participates in collective rituals with Priests |

| Ritualist Type | Principal Magico-Religious Activity | Selection and Training | Social and Political Power | Professional Characteristics | Motive and Context |
|---|---|---|---|---|---|
| Medium | Healing and divination. Protection from spirits and malevolent magic. Agricultural rituals. Propitiation. | Spontaneous possession by spirit. Training in practitioner group. Ceremony recognizes status. | Low socioeconomic status. Informal political power. May designate who are sorcerers and witches. Exclusively moral. | Predominantly female; male secondary/rare. Part-time. Collective group practice. | Acts primarily for clients at client residence. Also participates in public ceremonies. |
| Priest | Propitiation and worship. Protection and purification. Agricultural planting and harvest rites. Socioeconomic rites. | Social inheritance or succession. Political action. Incidental training and/or by group. Ceremony recognizes status. | High social and economic status. Political, legislative, judicial, economic, and military power. Exclusively moral. | Exclusively male. Full-time. Hierarchically organized practitioner group. | Acts to fulfill social functions, calendrical rites. Public rituals. |
| Sorcerer/Witch | Malevolent acts. Kill friends, enemies, neighbors, even kin. Cause illness, death, and economic destruction. | Social labelling/accusation. Attribution of biological inheritance. Innate abilities, self-taught or learned. | Low social and economic status. Exclusively immoral. May be killed. | Male and female. Part-time. Little or no professional organization. | Acts at client's request or for personal reasons such as envy, anger, jealousy, greed or revenge. Practices in secrecy. |

| Ritualist Type | Supernatural Power/Control of Power | ASC Conditions | ASC Techniques and Characteristics | Healing Concepts and Practices | |
|---|---|---|---|---|---|
| Shaman (Forager Shaman) | Animal spirits, spirit allies. Spirit power usually controlled. | ASC in training and practice. Soul flight/journey, death-and rebirth, animal transformation. | Isolation, austerities, fasting, chanting, singing, drumming and dancing. Collapse/ Unconsciousness. | Soul loss, spirit aggression, sorcery. Physical manipulations, sucking, blowing massaging and extraction. Plant medicines. | |
| Shaman/Healer (Agricultural Shaman) | Animal spirit allies and impersonal power (mana). Spirit control, spells, charms, exuvial and imitative. Power controlled. | ASC in training and practice. Shamanic and mystical ASC. Some have soul flight, animal transformation. | Isolation, austerities, fasting, chanting, singing, drumming and dancing. Collapse and unconsciousness. | Extraction and exorcism, countering spirit aggression. Physical manipulations, massage. Plant medicines. | |
| Healer | Superior gods and impersonal power (mana). Charms, spells, rituals, formulas and sacrifice. Propitiate & command spirits. | ASC induction limited. No apparent ASC. | Social isolation; fasting; minor austerities; limited singing, chanting or drumming. | Exorcism and prevent illness. Physical manipulation of body, empirical medicine, imitative and exuvial techniques. | |

**Table 1.** *Cont.*

| Ritualist Type | Supernatural Power/Control of Power | ASC Conditions | ASC Techniques and Characteristics | Healing Concepts and Practices |
|---|---|---|---|---|
| Medium | Possessing spirits dominate. Propitiation and sacrifices. Power dominates, out of control, unconscious. | ASC in training and practice. Possession ASC. | ASC induced through singing, drumming, and dancing. Tremors, convulsions, seizures, compulsive behavior, amnesia, dissociation. | Possession and exorcisms. Control of possessing spirits. |
| Priest | Power from ancestors, superior spirits or gods. Impersonal power and ritual knowledge. Propitiation and sacrifices. No control over spirit power. | Generally no ASC apparent or very limited. | Occasionally alcohol, sexual abstinence, isolation, sleep deprivation. | Purification and protection. Public rituals and sacrifices. |
| Sorcerer/Witch | Power from spirits and ritual knowledge. Contagious, exuvial and imitative magic, spells. Power can be unconscious, out of control. | Indirect evidence of ASC in reported flight and animal. transformation. | Night-time activities. | Illness by consuming victim's soul, spirit aggression, magical darts that enter victim, unconscious emotional effects of envy, anger, etc. |

### 2.1. Social Predictors of Ritualist Types

Winkelman's (2022) analyses[6] of subsistence and social variables associated with each ritualist type illustrate the source of some of the significant differences among them and a model of sociocultural religious evolution. This involves transformation of primordial foraging ritualists (Shamans) found worldwide in the premodern foraging and horticultural societies through effects of intensive agriculture, warfare and political integration. These specific social effects on religious evolution are illustrated in the following distinctive ecological, productive and social relations associated with each type of ritualist:

- Shamans (Foraging Shamans), the only ritualists in societies with a principal reliance on foraging and without intensive agriculture, supra-community political integration or warfare;
- Shaman/Healers (Agricultural Shamans) found in societies with intensive agriculture but lacking supra-community political integration, and generally with the presence of another ritualist, the Priests;
- Priests are found in intensive agricultural societies with supra-community political integration and are always found in societies with the following types of ritualists:
- Healers, who are generally found in agricultural societies, but significantly predicted by supra-community political integration and the practice of warfare for resources;
- Mediums, characteristic of complex societies and significantly predicted by supra-community political integration and the presence of warfare for captives; and
- Sorcerer/Witches in societies with intensive agriculture and supra-community political integration, but lack community integration (see Winkelman 2022 for analyses).

Since these ecological and social variables were not used to determine the practitioner types, the relationships provide an independent confirmation of the validity of the ritualist types presented (see Winkelman 1986, 1992 for confirmatory analyses). These ritualist types are found in specific configurations related to subsistence and political conditions, and as illustrated in Figure 1, present a model of sociocultural religious evolution (also see Winkelman 2022).

**Ritualist Types and Configurations in Relation to Subsistence and Sociopolitical Conditions**

| 1 Ritualist Type | 2 Ritualist Types | 3 Ritualist Types | 3 Ritualist Types | 4 Ritualist Types |
|---|---|---|---|---|
| | Priest | Priest | Priest | Priest |
| | | | Sorcerer/Witch | Sorcerer/Witch |
| | | Healer or Shaman/Healer | Healer | Healer |
| Shaman or Shaman/Healer | Shaman/Healer or Medium | Medium | | Medium |

**Subsistence and Sociopolitical Conditions**

| | | | | |
|---|---|---|---|---|
| No Political Integration | Local Political Integration | Hi Political Integration | Hi Political Integration | Hi Political Integration |
| No External Warfare | | Warfare- Captives | Judiciary | Social Stratification |
| No Intensive Agriculture | | War--Non-land Resources | Low Community Integration | |
| Foraging | Intensive Agriculture | Intensive Agriculture | Intensive Agriculture | Intensive Agriculture |

**Figure 1.** Ritualist Types, Configurations and Sociocultural Evolution.

*2.2. Cross-Cultural Features of Foraging Shamans (SCCS Variable 879)*

The ritualists in the group (cluster) labeled as (Foraging) Shamans shared the following characteristics, constituting an empirically derived and cross-culturally valid set of features of Shamans:

- Pre-eminent group leader who performs a dramatic night-time communal ritual involving enactments, drumming, dancing and singing;
- Principal ritual functions of spirit communication for healing, divination, hunting and sorcery;
- Selection for the role through spirit encounters interpreted as an illness and experienced in visions and dreams;
- Training with vision quests in the wilderness with fasting, austerities and often psychoactive plants;
- An initiatory experience of death by animals which killed and dismembered the initiate, followed by a rebirth and a reconstruction by animals incorporated as a principal power;
- Ritual preparations of fasting and sexual abstinence;
- Altered states of consciousness (ASC) conceptualized as magical or soul flight (out-of-body experience) and an experience of personal transformation into an animal, but notably the absence of possession in the ASCs;
- Healing practices focused on recovery of patient's lost soul, combating evil spirits and the extraction of magical darts causing illness;
- Causing illness and death through darts, sorcery and soul theft; and
- Directing hunters and calling animals.

*2.3. Biogenetic Bases of Shamanism*

Support for using this empirically derived cross-cultural pattern to determine what is a shaman is further supported by the correspondences of these features with biologically-based functions that produce these common features. These correspondences have biological bases in the phylogenetic origins and evolution of hominin ritual as a community integration process; mimetic and other ritual effects producing the physiological dynamics of ASC and healing; and central features of spirits and animal powers as personal and social identity, features directly related to innate operators (modules) of the evolved psychology of hominin adaptation (Winkelman 2009, 2010a, 2010b, 2021b). These biological

bases producing worldwide uniformities in foraging ritualists (Shamans) derive from the following biogenetic structures:

- A collective night-time/overnight conspicuous display with community drumming, dancing and singing which have deep evolutionary antecedents illustrated in the homologous sociality-enhancing maximal displays of chimpanzees (Winkelman 2009, 2021b);
- Selection for the shamanic role based on spontaneous visions, dreams and sickness involving natural tendencies for ASC that enhance access to and integration of unconscious processes (Winkelman 2010a, 2011, 2021c);
- Training in the alteration of consciousness (i.e., ASC) induced by practices of isolation in the wilderness, fasting, abstinence and austerities that stimulate the neuromodulatory neurotransmitter systems (Winkelman 2017);
- ASC induced by engaging the mimetic operator (dancing, singing, drumming) which produce an activation of the endogenous opioid system (Winkelman 2017, 2021a);
- Ritual activities leading to exhaustion and collapse, producing experiences of communication with spirits and out-of-body experiences reflecting innate modules (Winkelman 2015, 2021c);
- Initiatory experiences of death/dismemberment from attacks by animals and a rebirth that produces experiences of personal transformation and of incorporating animal powers into identity and basic structures of self-consciousness (Winkelman 2010a);
- Spiritual experiences produced by stimulation, integration and dissociation of innate modular cognitive structures operators (Winkelman 2021d); and
- Healing by recovery of lost soul, extraction of objects and removal of sorcery by ritual elicitation of endogenous healing mechanisms (Winkelman 2010a)

The congruences of these shamanic features with features and functions of humans' evolved psychology illustrate they are not arbitrary cultural features but have biogenetic bases; consequently, they show these variables are the most objective criteria for determining an etic transcultural characterization of shamans in comparative perspective. These provide five major biogenetic aspects for the bases of shamans:

- Mimetic community ritual: mimetic enactments in dramatic performances, collective drumming, dancing and singing;
- Traumatic selection: a psychological dynamic manifested in spontaneous visions, special dreams and psychosis-like sickness;
- ASC: out-of-body (soul flight) experiences and initiatory experiences of death and dismemberment by animals followed by rebirth;
- Animal powers: Central roles of animals in formation of personal powers and experiences of personal transformation into animals;
- Healing: involving recovery of soul loss or theft, extraction of sorcery objects and removal of effects of attacking spirits

These biogenetic bases indicate the presence of a shaman is a justified assumption for ancient foraging and simple agricultural societies worldwide. But agricultural intensification produces global changes in the dynamics of societies and religion, with a new form of ritualists in the role of the Priest who predominates with the emergence of supra-community political integration. Nonetheless, the core features of Foraging Shamans are initially maintained in Agricultural Shamans such as healing rituals with drumming and singing; selection by spirits in visions, dreams and illness; training and practices with ASC and experiences of soul flight, animal transformation and death and rebirth; powers derived from animals; and healing practices related to soul loss and spirit aggression.

But intensive agriculture subsistence produces new patterns of ritual behavior manifested in Priests who now lead community-wide rituals, relegating the Agricultural Shamans to private rituals with clients and their families. Agricultural Shamans also typically have professional organizations that provide training and recognition, features reflecting the social complexity and population concentrations enabled by large sedentary communities. Greater complexity of society also supported a role specialization among Agricul-

tural Shamans (i.e., just diagnosis, treating specific kinds of illness or agricultural rites) that would reduce intra-group competition among the practitioners.

Eventually, other ritualists—Healers, Mediums and Sorcerers/Witches—also emerge in societies with intensive agriculture and hierarchical political integration and are associated with war for resources and war for slaves and captives. These represent dynamics that may have led to the transformation and eventual demise of an ancient Chinese Foraging Shaman and the emergence of new types of ritualists illustrated in the distinctive features of various historical *wu*.

### 3. Comparative Analyses: Different Types of *Wu* in Cross-Cultural Perspective

From the earliest recorded periods, literary sources identify distinctly different types of *wu* rather than a homogenous profile for all practitioners called *wu*. Tong (2002) reviews the evidence for both pre-historic (Neolithic) and historic *wu* and references anthropological theories of religious evolution describing three subsequent types of ancient Chinese *wu* ritualists: magicians (including shamans); priests; and sorcerers and witches. Literary sources from the Shang period reveal a further early differentiation between female religious practitioners called *wu* and male ones called *xi* (覡 *wu xi* and 巫醫 *wu yi*) (see Cai 2014; Lin 2009). This followed a division of labor between men's involvement in government, exemplified in the King's (*wáng* 王) role; and women's role in religion as diviners on the King's behalf, a function that was partially usurped by the male ritualists who controlled them. Lin further distinguished *wu yi* from other ancient *wu* called "commoner shamans" (民巫 *Min wu* (Lin 2016)), a distinction paralleled in Michael's (2015) contrast of two different forms of early *wu*, noting "two separate traditions of early Chinese shamanism that I later call bureaucratic shamanism and independent shamanism . . . [the latter a] tradition of folk shamanism that only tangentially relates the *wu* to official positions of rulership and bureaucratic institution" [pp. 652–53]). While independent shamanism resisted or even rejected central authority and operated independently of the priests and rulers, the bureaucratic shaman was subordinated to centralized authority and "all other functionaries of the bureaucratic structures of state religion, including priests, temple officers, sacrificers, diviners, and scribes" (p. 671). But eventually, *wu* became depreciated and even criminalized, as their practices came to be characterized as sorcery and practitioners were subjected to execution, exemplified in the wū gǔ (巫蛊) described by Lin (2009) and Cai (2014).

These overlapping distinctions illustrate a broad consensus that from the beginning of Chinese religion, there were several distinctive types of *wu* ritualists. My analysis focuses on these distinctive types of *wu* ritualists across the eras of ancient China, oriented by information from Lin (2009); Tong (2002); Cai (2014) and Sukhu (2012) who provide similar distinctions about these types of early *wu*:

(1) Pre-historic Neolithic *wu* revealed in archaeological (Tong 2002) and linguistic (Hopkins 1945) evidence;

(2) Ancient *wu* (focused on the men *wy xi*, rather than women *wu*) from the Shang and Zhou periods;

(3) Commoner *wu* (Lin 2009), called community *wu* by Tong (2002), is a professional class recorded in the late Zhou period, particularly in the south, and strongly contrasting with the bureaucratic practices of the state *wu*; and

(4) State religious officials, the official *wu* (*siwu* (司巫), *nanwu* (男巫), *nüwu* (女巫) and others of the Zhou dynasty (1046–256 BC) described by Lin (2009); Michael (2015) and Tong (2002), who labels them as priests.

There are also other forms of *wu* described for the Han and Qin periods:

(5) Female *wu* (巫尫 *Wu Wang* [Du and Kong (2000)] and 巫嫗 *Wu Yu* [Takigawa (2015)]) of the Warring States period (from Cai (2014) and Sukhu (2012)); and

(6) Wū gǔ (巫蛊) (Lin 2009; Cai 2014) of the Han period.

Two contemporary examples of ritualists alleged to be Chinese shamans are also reviewed to illustrate the continued use of the shaman concepts for Chinese ritualists and their variable correspondences with the etic model:

(7)　the Chinese Reindeer-Evenki, whose indigenous ritualists, the *šaman*, are described by Heyne (1999); and

(8)　the *bo* of the Tu ethnic group of Qinghai Province in Northwest China, who is called a shaman by Xing and Murray (2018).

The characters representing prehistorical *wu* (Figure 2), healing *wu* (Tables 2 and 3) and official *wu* (Table 4) illustrate a range of representations. The features of these different types of *wu* are presented in Tables 3 and 5 and in the following material, where they are compared with the etic ritualist types identified by Winkelman (1986, 1990, 1992, 2010a; also see Table 1). In the Tables 3 and 5, the initial letter of the etic ritualist types (i.e., S, AS, H, M, P, S/W) is assigned to those characteristics of Chinese ritualists who exhibit characteristics unique to an etic type (i.e., S for Shamans' animal powers, soul journey; M for Mediums' possession; and H, P for Healers' and Priests' formal political and judicial powers).

*3.1. Pre-Historical Wu*

There is linguistic, archaeological and mythic evidence of the presence of an ancient Chinese *wu* (Tong 2002) that has central characteristics of Shamans. These ancient *wu* practices, referenced in representations of *wu* characters in Shang and Early Chu dynasty records, are interpreted by Hopkins (1945) as depicting "the unmistakable shape of the dancing Shaman" (p. 3; see Figure 2). Hopkins (p. 4) asserts the religious centrality of these actions of a shaman dancer represented in the *ku wén* character ( 爽 ), concluding that *ku wén* and *wu* represent the same sound and word. Furthermore, they resemble representations depicted in "early Bronzes and the inscribed Bones of the Honan Find". Hopkins suggests these Lesser Seal sources present a recognizable portrait of a dancing or posturing figure, a direct mimetic representation using straight and curving lines to symbolically represent an unmistakable dancing figure (see Figure 2). These dancing activities have a direct affinity to the core biogenetic mimetic bases of the practices of shamans.

### Wu Characters (Hopkins)

| | |
|---|---|
| 巫 | Shuo Wên Lesser Seal form |
| 巫 | *Wu*, shaman |
| 無 | *Wu*, the negative verb |
| 舞 | *Wu*, to posture |
| 無　舞 | *Wu*, to dance (ancient and modern forms) |
| 爽　爽 | Shuo Wen Lesser Seal versions of wu (negative verb) with feathered plumes or plants suspended from arms of dancer |

**Figure 2.** Characters Representing Prehistoric Wu.

In Shang and Early Chu dynasty records, the *wu* character with meanings of "to posture" and "to dance" is found in both the Kangxi Dictionary and the Shuo Wen of Hsu Shen. Hopkins identified the earlier form of representation for the *wu* in the Shou Wen which he says defines the meaning of *wu* (*wu chu yeh* 巫 祝也) as "Invoker" or "Imprecator" (p. 3). Shamans' activities are central to the meanings assigned to this character: "invoker"—with meanings of cause, conjure and incantation—as well as those of "imprecator"—meaning to evoke evil and curse, as well as to pray and ask for and entreat. These vocal activities reflect

the shamans' chanting and singing challenges to the spirits and seeking their removal—or sending them to enemies.

Hopkins proposes that the lower component was not originally *kung* (early form 廾; Lesser Seal form 𠬞), referring to two hands, but rather the character *ch'uan* (舛), referring to two feet (p. 4). Either interpretation confirms the mimetic base in the clapping or dancing figure of the shaman, activities that can induce ASC and are a core feature of shamanism. Hopkins reviews evidence showing that the meanings of the character *wu* are in agreement with the notion of dancing represented in *ts'ung wu chih i*, (從 舞 之 意) referring to the ritual dancing of the typical *yü* (雩) ceremony customary in the Yin Dynasty.

Early Chou texts represent *wu chung* (無 終 or *wu tung* 無 冬) graphically as meaning "to dance" but functionally, its representation as *wu* has the meaning of "the negative verb". Hopkins proposes that a disentanglement of the three modern characters that are pronounced as *wu* can be achieved by reference to their primitive contours that reveal their pristine meanings. Hopkins proposes the three modern characters represented in English as *wu* have interrelated meanings of "shaman", "the negative verb" and "to posture," and that "all be traced back to one primitive figure of a man displaying by the gestures of his arms and legs the thaumaturgic powers of his inspired personality" (p. 4).

Tong (2002) integrates myth, legend and history with archaeological findings to illuminate the nature of these ancient Chinese ritualists. Tong calls attention to the shamanic significance of drums excavated from a cemetery in Shanxi Province in tombs associated with the Longshan (Lung-shan) period (3000–1900 BC). These pottery drums, several covered with alligator skin and found in association with musical stones and other high-status items, illustrate the ancient association of the *wu* with classic Shaman features of music, dancing and drumming. Tong describes a find of Neolithic pottery bowls that represent a group of dancing figures with depictions of tails which he suggests represents the intimation of animals in a ceremonial dance, another core feature of the Shaman's relationships to animals. A compilation of documents, "The Tongdian," provided similar evidence of this association of pre-historic *wu* with animals illustrated by groups of dancers wearing wooden masks depicting animals (dog and pig). Tong suggests that *wu* activities were represented in many of "the drums, chime stones, whistles, and flutes discovered at various Neolithic sites in China" (p. 51). Tong (2002, p. 52) notably emphasizes a feature core to shamanism: "In all these ceremonies, the drum was the most significant instrument."

These analyses of linguistic and archaeological data provide evidence of ancient Chinese traditions representative of a Foraging Shaman, practices which existed before the differentiation of the functions of the tribal *wu*, before written records and the development of "shaman-king" and a state *wu*. The principal biogenetic aspects of Shamans outlined above are present in the limited evidence about their central societal role in collective rituals; the mimetic complex of dancing, drumming and singing; and the significance of animals, including animal transformation. The available data does not provide a full profile of the etic Shaman but provides confirmatory evidence of the presence of features of the Shamans found in the ethnological research, confirming that *wu* shamans existed in Chinese pre-historic antiquity.

**Table 2.** Characters Representing Types of Healing Wu.

| Chinese Character | Pinyin | English |
|---|---|---|
| 巫 | *Wu* | |
| 巫 祝也 | *Wu zhu ye* | Invoker, imprecator (Hopkins 1945); Sorcerer (Shuowen Jiezi 2014) |
| 覡 | *Wu Xi* | Male Shamans (Cai 2014); Male Sorcerer (Xu 2002) |
| 巫醫 | *Wu Yi* | Physicians who treat medical and surgical conditions (Lin 2009) |
| 民巫 | *Min Wu* | Commoner *Wu* (Lin 2009, 2016) |
| 官巫 | *Guan Wu* | Generic term for officially appointed *Wu* (Lin 2016) |
| **Female Wu** | | |
| 巫尪 | *Wu Wang* | Female witch (Du and Kong 2000) |
| 巫嫗 | *Wu Yu* | Female witch/sorcerer (Takigawa 2015) |
| 巫蠱 | *Wu Gu* | Witchcraft activities (Lin 2009; Cai 2014) |

*3.2. Ancient Wu Xi*

A variety of forms of healing *wu* are reported (see Tables 2 and 3 for characteristics). Literary sources from the Shang period reveal an early distinction of male religious practitioners called *wu xi* (Cai 2014; Lin 2009). Lin characterizes the male *wu xi* of the pre-Qin and Han periods as "ancient shamans." These ancient *wu xi* (paraphrased from Lin (2009)) had central functions of healing and divining, as well as performing sacrifices to the gods and spirits and rites of ancestor worship, assisting the sovereign in mourning rites. *Xi* were people of high status and superior intelligence who held knowledge, respect and high regard. They were hereditary aristocrats who exhibited correct demeanor, loyalty and trustworthiness and brought glory to the Kings. Their societal power was illustrated in overseeing the religious affairs of the ruling class and the entire society (state) and they enjoyed relatively great influence. Their rituals involved special sacrificial vessels, vestments and the use of statues, animal figures and representations of gods and ghosts, using sacrifices, incantations and prayers to gods to obtain blessings. The ancient *xi* likely engaged in ASC as suggested by their use of visions to illuminate matters and their knowledge of how to "ascend and descend," suggesting the shamans' flights to lower and upper worlds. Their rituals involved preparation with special baths and fasting and ceremonies involving beating the drum, striking the bell and hollering to excite the heart, with vigorous steps and dance to stir up the energies. The healing functions of the *Wu xi* were considered their most important role, involving the use of drugs and plants to drive away pestilence, incantations for the removal of illness and exorcisms and blessings to avert misfortune.

**Table 3.** Comparisons of Shamans, Ancient Xi, Commoner Wu, Healers and Evenki *šaman*.

| Ritualist Type | Principal Magico-Religious Activity | Selection and Training | Social and Political Power | Professional Characteristics | Motive and Context of Ritual |
|---|---|---|---|---|---|
| Shaman (Forager Shaman = S; (AS for added Agriculture Shaman feature) | Healing and divination Protection from spirits and malevolent magic. Hunting magic (S) Cause illness, death (S, S/W) Assist Priests in agricultural rituals (AS) | Dreams, illness, and signs of spirit's request (S) Vision quest by individual alone in wilderness (S) Group training (AS). Ceremony recognizes status (AS) | High social status. Charismatic (S) Communal and war leader (S) Makes sorcery accusations. Informal power Moderate judiciary decisions (AS) | Predominantly male, female secondary (S) Part time. Individual practice (FS) Group ceremonies (AS) Specialized role (AS) Ambiguous moral status (S) | Acts at client request in community-wide ceremony Ceremony over-night (S) Performance in client group at client request (AS) |

| Ritualist Type | Principal Magico-Religious Activity | Selection and Training | Social and Political Power | Professional Characteristics | Motive and Context of Ritual |
|---|---|---|---|---|---|
| Xi-"Ancient Shamans" (Shang and Zhou periods | Healing & divining Divine people's fate, regarding illness Sacrifices for deceased, spirits and gods (M, H) Ancestor worship (P) Harvest, livestock (H, P) | Hereditary aristocrats (P) | Ruling class (H, P) Religious affairs of state (H, P) Very important and great influence | Males (H, P) Correct demeanor, the value of loyalty and trustworthiness People of superior quality, with high intelligence and respected (*S) | Religious functions of government (H, P) |
| Commoner Domestic Professional *wu* Shang and Zhou periods | Healing and protection Arts of divination Determine causes of misfortune Rituals to harm people and gain advantage (S) Worship ghosts, animals and natural phenomena | Hereditary family trade Innate selection or from disease (S, M) No formal group (S) | Commoner class (M) Familial/tribal function (S) Prestige but little power (S, M) Influence communal decisions, warfare and hunting (S) May be sanctioned, executed (SW) | Male and Female (S) Made a living but part-time (S, M) Often defective in body or handicapped (S*) Both heal and harm (ambiguous moral status) (S) | Services for clients Covet goods and heat people Profession for gain (H) |
| Healer (H) Etic Ritualist Type | Healing and divination. Agricultural and socioeconomic rites (H, P) Propitiation (H, P) Life-cycle rituals. | Voluntary selection, large payments to trainer. Learn rituals and techniques Ceremony recognizes status. | High socioeconomic status. Judicial, legislative, and economic power (H, P) Denounce sorcerers. | Predominantly male (H, P) Full-time (H, P) Collective training, practice and ceremony. Highly specialized role Predominantly moral | Acts at client request and treatment in client group. Collective public rituals with Priests |
| Chinese Reindeer-Evenki *šaman* | Treatment of illnesses Ward off misfortune Seasonal celebrations Life-cycle (marriage, funerals and memorials) Divining and predicting Malevolence (S, S/W) | Dreams and visions Illness/hysteria (S, M) Could be inherited Solitude and fasting in wilderness (S) Self/Spirit taught (S) Death-and-rebirth (S) | Great significance for community = charismatic (S) Unofficial leader (S) Not official clan head or political leader No material advantage | Male & female (S) Part-time More mentally capable(S) Courageous & strong Ambiguous morality—required altruism but might abuse power (S) | Altruism for benefit of community (S) Acted when person or community was disturbed Community obliged the ritualist to perform (S) |

| Ritualist Type | Supernatural Power/Ritual Techniques | ASC Conditions | ASC Characteristics and Techniques | Healing Concepts and Practices | |
|---|---|---|---|---|---|
| Shaman (Forager Shaman = S) (AS Additional Agriculture Shaman features) | Animal spirits & allies (S) Spirit power usually controlled (S) Impersonal mana (AS) Spells, charms, exuvial and imitative techniques (AS). | ASC in training and practice. Soul flight/ journey (S) Death-and rebirth (S) Animal transform (S) Mystical ASC (AS) | Isolation, austerities, fasting, chanting, singing, drumming and dancing. Collapse during ritual Unconsciousness (S) | Soul loss (S) Spirit aggression, sorcery. Sucking, blowing, massaging and object extraction (S) Plant medicines. Incantations for removal | |
| Xi-"Ancient Shamans" (Shang and Zhou period) | Animals (S?) Sacrifices (H, P) Pray for blessing (H, P) Use statues and icons (H) Incantations Bone Oracles (H) | Knew how to ascend and descend (S?) Vision illuminated matters and hearing penetrated them (S?) | Bathe and fast Beat drum, strike bell and holler Excite the heart, body Music, dancing, and drumming | Knowledge of plants Removal of illness Exorcism (H) Pray to gods for blessing to averting misfortune (H, P) Dream interpretation | |

**Table 3.** *Cont.*

| Ritualist Type | Supernatural Power/Ritual Techniques | ASC Conditions | ASC Characteristics and Techniques | Healing Concepts and Practices |
|---|---|---|---|---|
| Commoner Domestic Professional *wu* Shang and Zhou Periods | Spirits and ghosts of dead, not "orthodox gods" Incantations, blessings, praying (H) Curses, charms (AS) Sacrifice (H, P) Animals (?) (S) | Communicate with spirits Imaginary travel around heaven & earth (S) | Drumming and music performed by practitioners Purgative drugs (S*) | Expelling baleful influences, Exorcism (H, M) Pray to invoke the ghosts and gods for blessings (H, P) Sacrifice (H, P) |
| Healer Etic Ritualist Type | Superior gods and impersonal power (mana). Charms, spells, rituals, formulas and sacrifice. Propitiate & command spirits. Material divination system | ASC induction limited No apparent ASC (H, P) | Social isolation; fasting; minor austerities; limited singing, chanting or drumming. | Exorcism and prevent illness. Manipulate body Empirical medicine Imitative and exuvial techniques. |
| Chinese Reindeer-Evenki *šaman* | Supernatural qualities of animals (S) Changed themselves into animals (S*+) Master of the spirits who were subservient (S) | Techniques of ecstasy Soul travel to spirit world (S) Death-and-rebirth (S) Change into animal (S) | Drumming and singing central elements for trance Unconscious trance (S) May use alcohol (P*) | Soul loss/recovery (S) Remove malevolent influences of spirits Drum and singing had hypnotic influence on patient Medicinal herbs and plants |

Letters indicate contrastive features of practitioner type: S = Shaman feature (FS & AS), AS = additional Agriculture Shaman feature, H = Healer feature, P = Priest feature; S/W = Sorcerer/Witch feature; ? = questioned by authorities; * = often attributed to ritualist type but not part of etic model.

### 3.2.1. Comparisons of Ancient Wu Xi and Shamans (SCCS Variable 879)

Table 3 provides the characteristics of the *Wu xi* and other healing *wu*. The *xi* did have healing as a principal function, including the removal of illness characteristic of Shamans, but their healing practices of exorcisms and blessings reflect Healers. The ancient *wu* acquired their power through formally learned prayers and rituals rather than selection by the spirits and spontaneous and deliberate ASC characteristic of Shamans. The activities of ancient *wu xi* were based in ritual sacrifices and manipulation of ritual objects dedicated to the ancestors and gods of rain rather than the animal powers of Shamans. *Xi* notably lack central qualities of Shamans such as malevolent acts and hunting rituals; selection through illness and training alone in the wilderness; the focus on patient's needs rather than state functions; extensive ASC experiences involving soul flight/journey, death-and rebirth and animal transformation; and healing of sorcery and soul loss through a soul journey.

The *xi* did maintain some of the Shaman's core ASC activities in their engagement with practices of fasting, rituals involving beating the drum, striking bells and hollering, as well as activities that excite the heart and exercise the body, such as dancing. But despite statements about flight into the heavens and the mingling of divine and human entities suggested in some texts about *wu*, these accounts do not have contexts where they attest to experiences of ecstatic flights or out-of-body experiences characteristic of shamans (Keightley 1998). Keightley proposes instead that the spirits did not descend into or enter the *wu* (*xi*), but that instead they were descending to receive the sacrifices that were offered. In an analysis of ancient texts on the *xi*, Liu (2022, P 10 of 14) notes that "If we emphasize that ecstasy is a judgment criterion for a shaman, *Wu* and *Xi* cannot be categorised as shamans because this chapter did not mention ecstasy at all."

As Boileau (2002) noted, while ancient (archaic) *wu* and Siberian hunting shamans shared concerns with the well-being of the community, the *xi*'s involvement in the religious functions of sacrifices for the state and ancestors sharply contrast with Shamans' primary

role in communal healing. Similarly, the *xi*'s use of sacrifices, praying to gods for blessings and uses of statues and representations resemble Healers and Priests, not Shamans (See Table 3). While *xi* shared Shamans' healing features with the use of plant drugs for the treatment of illness, they differed in an emphasis on exorcism and praying to gods for blessings to avert misfortune. While some Shaman healing features related to soul loss that are typical of Siberia and the etic model are attributed to *xi*, closer examinations reveal that it is only a superficial resemblance. Williams (2020, p. 191) reviews a poem called "Summons to the Soul" from the Elegies of Chu reflecting activities involving shamanic concepts of soul recovery. The poem describes the ritual activity of summoning a soul back to save the patient's life, with chants designed to impede the soul's travel to heaven, the underworld or four directions. Williams notes, however, that this concept of a soul that can leave the body and the practices of soul loss recovery expressed do not involve the personal soul journey recovery involving ecstatic flight to the spirit world which is characteristic of Siberian practices and Shamans' soul recovery. Michael Puett (2002, cited in Michael 2015) illustrates central features of shamanism were absent even in early China, which was lacking the beliefs in three spiritual realms and the connecting axis mundi which were central to Eliade's characterizations of the central features of Siberian shamanism. Notably lacking in the ancient *wu* are Shamans' trauma and illness-related selection procedures and training involving ASCs (i.e., soul flight, death and rebirth and animal transformation). *Xi* also lack special relations with power animals, healing of soul loss and sorcery.

### 3.2.2. Comparisons of the Ancient Xi Wu with the Healer (SCCS Variable 881)

Healers provide healing and divination as principal functions, offering individual healing services rather than collective healing ceremonies. The Healers also participate in activities involving collaboration with Priests in collective agricultural rites and rituals of propitiation of gods. Healers also have an important function in officiating life-cycle rituals (i.e., naming, marriage and funerals). Healers acquire their roles through payments for training by other Healers, and when completed, a professional group certifies their status with a ceremony recognizing their professional status. Their activities as full-time specialists earn them a lucrative income. Healers are normally only males of high social and economic status and participate in political, legislative and judicial processes.

Healers' supernatural power comes from ritual knowledge, particularly the creation of charms and knowledge of spells and ritual techniques such as incantations and sacrifices used to propitiate the spirits. Their rituals do not seem intended to produce ASCs in patients. Nonetheless, clients may experience minor ASCs because of placebo and other consciousness-altering effects of the elaborate rituals, recited spells and incantations. In the divination of information, the Healers do not rely on ASC experiences but instead manipulate material systems (i.e., like I Ching or Tarot cards) that they randomize to produce patterns that they interpret for diagnoses. Exorcism, removing spirits that afflict or possess the patient, is the central healing ideology of Healers and Mediums. Healing rituals involved the repeated recitation of spells that appear to have hypnotic effects and powerful encouragements apparently intended to have suggestive and placebo effects. Although the Healers did not emphasize obvious ritual ASC, these are nonetheless suggested by their training and pre-ritual periods of social isolation and fasting and limited singing, chanting and percussion.

*Xi* features also characteristics of Shamans involving ASC induction (beating the drum and vocalizations) and healing through plant medicines and expelling bad influences—these are not unique to Shamans but are shared by Healers as well. *Xi* more closely resemble the Healers of Winkelman's etic model, sharing characteristics such as the following: religious functions such as propitiation of deities (King's ancestors); social qualities such as high socioeconomic status and governmental powers; being only males who engage in functions with priests for collective rituals; providing sacrifices to appease superior gods; seeking blessings/propitiation; and healing through exorcism and the prevention of misfortune. The Ancient *xi* use of oracle bones, a material system for divinatory purposes, reflects

typical divinatory practices of Healers (use of mechanical systems rather than intuition), not Shamans.

### 3.3. Commoner Wu

Commoner *wu* (*Min wu* 民巫 Lin 2016) were members of a professional class of the late Zhou period. Tong (2002, pp. 62–65) summarizes the characteristics and functions of what he referred to as "community" practices corresponding to the commoner *wu*. The primary functions of these Commoner *wu* were providing healing services and prayers, blessings and invocations to protect from disasters. Commoner *wu* were also skilled in divination to determine causes of illness and other misfortune but notably lacked the formal ancestral worship system characteristic of state rituals. Instead, these "community" practices provided ceremonies for individuals and families with rituals involving the worship of ghosts, animals and natural phenomena and provided spiritual support, particularly to determine the client's fortune or misfortune. Commoner *wu* also performed rituals to harm people and gain advantage; consequently, the commoner *wu* were also seen as potentially dangerous figures that needed to be controlled, or even banned.

The Commoner *wu* frequently practiced an inherited family trade but were selected on the basis of innate abilities or as a consequence of disease, or were even bodily handicapped. They could be male or female and acted on a part-time basis in providing services at low expense to clients. Their functions were based on their abilities to communicate with the gods and other spirits whom they queried with divinatory rituals and addressed with sacrifices to obtain their favor. They used charms and spells, as well as prayers, exorcisms and sacrifices.

While their role included services as tribal functionaries, they were without political powers, formal organization or political privileges; nonetheless, they had prestige and influenced communal decisions regarding agricultural activities, as well as warfare and hunting. Their performances were conducted without written scriptures but with singing, drumming and dancing performed by the practitioners themselves. A significant service was exorcism rituals performed with incantations and sacrifices. Beyond singing, drumming and dancing, there is evidence of their ASC in purgative drugs used and reported ecstatic states of "imaginary travel around heaven and earth in chariots drawn by dragons and phoenixes" (Tong 2002, p. 64).

Tong concludes that, in contrast to the state *wu* religious officials of the Xia, Shang and Zhou dynasties, the southern areas persisted with the ancient traditions and that these "practitioners of southern folk religion still can be called *wu*" (p. 65). Tong also proposed other translations for these earlier ritualists as magicians, sorcerers and even physicians.

Comparing the Commoner Wu and Shamans (SCCS Variable 879)

A comparison of the characteristics of Commoner *wu* with Foraging and Agricultural Shamans (referred to here as Shamans) shows both similarities as well as some divergences. Sufficient detailed historical information is lacking for all relevant features of the Commoner *wu*, but they reflect Shamans in their principal functions such as healing services, divination and engaging in practices of sorcery. Commoner *wu* also exhibit Shamans' features in selection on the basis of innate abilities or as consequences of disease, as well as family tradition. The tribal and familial focus of Commoner *wu* is consistent with the Foraging Shamans' communal activities, as is their role as familial or tribal functionaries without political power, formal organization or special privileges but instead having prestige and influence in communal decisions.

But community-wide rituals are not reported for Commoner *wu*; instead, client-based practices characteristic of Agricultural Shamans are reported, as illustrated in their emphasis on economic gain from their clients. Commoner *wu* also perform agricultural rituals characteristic of Agricultural Shamans, as well as the worship of ghosts, animals and natural phenomena, rituals more typical of Priests, Healers and Mediums. Both Foraging Shamans and Agricultural Shamans' features are also present in the commoner *wu's*

part-time profession and reported involvement with activities of warfare and hunting and malevolent rituals. But it is Agricultural Shamans that attempt to harm through their use of curses, witchcraft (black magic) and charms, as do Commoner *wu*. Commoner *wu* performances conducted without written scriptures or temples but with drumming and music performed by practitioners themselves are consistent with, but not exclusive to, Foraging and Agricultural Shamans. The Commoner *wu*'s engagement with mimetic activities (singing, drumming and dancing performed by practitioners) are features which are widely shared by other etic types (Healers, Mediums). But there is a suggestion of uniquely Shamans' ASC in ecstatic states of imaginary travel around heaven and earth and even animal relations in the ecstatic travel via dragons and phoenixes and the worship of animals.

Like Foraging and Agricultural Shamans, commoner *wu* provided healing services; but while the healing functions of Commoner *wu* involves Shamanic concepts in expelling baleful influences and presumably countering sorcerers and witches, the Commoner *wu* also focus on exorcism, prayers, blessings and invocations to protect from disasters and sacrifices; these are not like Shamans' healing features, but instead reflect activities typical of Healers and Mediums. Notably lacking in Commoner *wu* are the Shamans' practices of the healing of soul loss. Furthermore, commoner *wu's* healing functions differ from Foraging and Agricultural Shamans' healing in the focus on spiritual support of the client's fortune or misfortune and their emphasis on economic gain from their clients.

While some aspects of Shamans' ASC are present, the typical experiences are not directly reported for Commoner *wu*, and notably absent are the death-and-rebirth experiences and the central role of animals as powers, identities and a form of personal transformation, characteristic of Foraging and Agricultural Shamans. Features more characteristic of Agricultural Shamans and Healers are seen in the Commoner *wu*'s involvement with agricultural rituals and charging for services to clients. Features of the Commoner *wu* such as agricultural rituals resulted from the effects of intensive agriculture, features also found in Agricultural Shamans and Healers who also assist the Priests in agricultural rituals. Similarly, the Commoner *wu*'s lower status likely reflects the higher status *wu* priests, a feature also exhibited by Agricultural Shamans. Techniques reported for Commoner *wu*—incantations, blessings, praying, curses and charms—appear to reflect the increasing commodification of healing practices manifested in the Commoner *wu's* focus on earning from clients. These more resemble features of Agricultural Shamans and Healers—ritual techniques involving spells and charms with exuvial and imitative rituals.

The Commoner *wu* far more closely resemble Foraging Shamans than did the Ancient *wu*. These similarities include the evidence of trauma and illness in selection; the presence of ecstatic ASC reflective of soul flight; the relationships to animals; and healing involving extraction of illness and addressing sorcerers. But notably, the Commoner *wu* lack a communal audience and instead perform agricultural rites, use ritual techniques of charms and spells, have lower social status and perform client-focused rituals, as do the Agricultural Shamans with which they most directly correspond.

Lin (2009) characterizes the appearance of commoner or domestic professional shamans in the literature as representing a major change. Lin attributes the appearance in the literature of information about Commoner *wu* as a result of the effects of the collapse of the Zhou feudal system (7th–6th centuries BC). But commoner *wu* practices are unlikely to have been a consequence of the decline of the feudal system, although it may have elevated their relative societal importance in the ensuing vacuum. If the Commoner *wu* did not exist in the broader society during the Shang and Zhou dynasties, from where did it originate? It was not a devolution of the bureaucratic *wu* or ancient *wu yi*, who had already long departed from the Foraging Shaman pattern. Rather, this appearance of the Commoner *wu* in the literature must reflect the emergence of record-keeping related to these ongoing activities that undoubtedly had continuities with the pre-historic *wu*.

### 3.4. Bureaucratic Wu Priests

While archaeological evidence and texts lack sufficient detail to infer all of the social functions and characteristics of the *wu* before the emergence of writing and the state system, evidence indicates an ancient differentiation of functions of the tribal *wu* in the development of "shaman-king" and a state *wu* (Tong 2002). Tong points out ancient changes in the functions of the *wu* in the emergence of a new polity formed through their symbiosis with the tribal elite, leading to a "religious elite of *wu* [who] reconstituted themselves and became priests" (p. 53). Tong (2002) reviews ancient texts on the consequences of these religious and political transformations during the emergence of chiefdoms ("Period of the Five Legendary Emperors", third millennium BCE) when functions of the new elite ritualist's centralized religious activity by organizing the worship of the ancestors of the ruling clan as gods of the whole polity. These bureaucratic functionaries of the government during the Xia, Shang and Zhou periods nonetheless had *wu* in their titles, along with other formal designations that revealed their distinctive roles as officials of the state (see Table 4 for characters).

**Table 4.** Characters and Names of Wu Officials.

| Chinese Character | Pinyin | English Terms Used by von Falkenhausen (1995) and Lin (2009) |
|---|---|---|
| 男巫 | *nanwu* | Male shamans |
| 女巫 | *nüwu* | Female shamans |
| 司巫 | *siwu* | Manager of the Spirit Mediums |
| 師巫 | *shiwu* | Officers |
| 巫師 | *wushi* | Instructors of Spirit Mediums |
| 巫恆 | *wuheng* | Spirit Mediums |
| 無數 | *wushu* | Male and Female Spirit Mediums |
| 宗 | *zong* | Temple Official |
| 祝 | *zhu* | Invocators |
| 大祝 | *dazhu* | Great Invocator |

Michael (2015, p. 685) reviews texts that illustrate the distinctive characteristics of these *wu* bureaucratic functionaries that he refers to as priests (*zhu* 祝) and temple officers (*zong* 宗). Michael (p. 685) characterizes the primary functions of these ritualists as involving the performance of rituals for the ancestors. And rather than seances or face-to-face communication with these spirits, the ritual performances involved sacrifices and other offerings, devotional prayers and songs and liturgical performances. Their religious functions are assisted by temple officers who manage ritual objects such as vessels and jade and administer the sacrificial offerings of animals and fruits of the harvest. Michael notes that with the institutionalization of these roles in activities of the ancient Chinese state religion within the spirit temples and ancestral halls that "the *wu* are no longer recognized: shamanic authority has given way to centralized authority" (Michael 2015, p. 685).

Tong characterizes the activities of these *wu* state religious officials as involving primary responsibilities for state affairs and the needs of the royal family. These full-time religious practitioners were formally appointed as government officials to exercise political powers and direct economic activities. Their positions depended on skills and knowledge from study and long-term training, not supernatural gifts. They held high prestige from their positions and were affiliated with important political groups. The state *wu* engaged in political and economic activity for the state, with each ritualist type holding specific and delimited responsibilities. Some controlled groups of mediums (also called *wu*) accompanied religious officials in dancing during the rainmaking rituals and addressed great calamities in rituals of crying, praying and singing. Their rituals followed clearly established practices dictated in written texts and performed in special temples on scheduled days. Their rituals emphasized sacrifices, with some divination (dream interpretation) and exorcisms.

**Table 5.** Comparisons of Priests, Mediums and Sorcerer/Witches with Tu *Bo*, State and Female *Wu* and Wugu.

| Ritualist Type | Principal Magico-Religious Activity | Selection and Training | Social and Political Power | Professional Characteristics | Motive and Context of Ritual |
|---|---|---|---|---|---|
| Priest (P) Etic Ritualist Type | Propitiation and worship. Protection and purification. Agricultural planting and harvest rites. Socioeconomic rites. | Social inheritance or succession. Political action. Incidental training and/or by group. Ceremony recognizes status. | High social and economic status. Political, legislative, judicial, economic, and military power. Exclusively moral. | Exclusively male. Full-time. Hierarchically organized practitioner group. High social and economic status. | Acts to fulfill social functions, calendrical rites. Public rituals. |
| Official *wu* (*siwu*, *nanwu*, *nüwu*)—Zhou dynasty | Sacrifices to gods in ancestral temple (P) Protect from disaster (P) Rituals for rain-making, driving away pestilence, protect harvest (H, P) Funerary services (H) | Selected/appointed by government personnel (P) State officials in charge of training (H, P) | Part of official structure and ruling circles (H, P) Regular members of the bureaucracy (H, P) Political and economic activity (H, P) | Male , full-time (P, H) Specialized hierarchy (H, P) Skills and knowledge from study High prestige from their positions (H, P) | Social functions-Formal rituals at temples (P) Calendrical rituals at temples at specified times (P) |
| *Bo* of the *Tu* (Xing and Murray) | Ensure good weather and agriculture (P, H) Ancestors worship (P) Life cycle events Community well-being Resolve conflicts Not a source of healing | Hereditary, passed down from one generation to the next within the same family (P) | Leadership in collective rituals (P) Intervene to resolve conflicts (P) | Only males (P, H) Part-time specialist | Annual festival (P) Organized by village association Public festival in temple (P) Entire community participates (S) |
| Medium (M) Etic Ritualist Type) | Healing and divination. Protection from spirits and malevolent magic. Agricultural rituals. Propitiation. | Spontaneous possession by spirit. Training in practitioner group. Ceremony recognizes status. | Low socioeconomic status. Informal political power. May designate who are sorcerers and witches. Exclusively moral. | Predominantly female; male secondary/rare. Part-time. Collective group practice. | Acts primarily for clients at client residence. Also participates in public ceremonies |
| Female *Wu*—Qin and Han dynasties, especially Warring States | Communicate with gods Assure well-being of king and state Control rains for agriculture (P, M) Healing rituals Funeral rites | Appointed to positions in the royal courts Group training (M) Male spirit or god descends on person (M) | Generally low status (M) Contributed to bureaucratic and political decisions with divination (H, P) | Female (M) Organization served the royal court (M) | Assure well-being of king and state (P) Regular times throughout year (P) |
| Sorcerer/ Witch (SW) Etic Ritualist Type) | Malevolent acts. Kill friends, enemies, neighbors, even kin. Cause illness, death, and economic destruction. | Social labeling/accusation. Attribution of biological inheritance. Innate abilities, self-taught or learned. | Low social and economic status. Exclusively immoral. May be killed. | Male and female. Part-time. Little or no professional organization. | Acts at client's request or for personal reasons such as envy, anger, jealousy, greed or revenge. Practices in secrecy. |
| *Wugu* | Cause illness and death (S, SW) | Denounced by government officials (SW) | None May be killed, executed (SW) Condemned as immoral (SW) | Males and females (S, SW) Lower class (SW) | Personal gain, payment (SW) Revenge (SW) |

**Table 5.** *Cont.*

| Ritualist Type | Supernatural Power/Control of Power | ASC Conditions | ASC Techniques and Characteristics | Healing Concepts and Practices |
|---|---|---|---|---|
| Priest (P) (Etic Ritualist Type) | Power from ancestors, superior spirits or gods. Impersonal power and ritual knowledge. Propitiation and sacrifices. No control over spirits. | Generally no ASC or very limited | Occasionally alcohol (P) Sexual abstinence, isolation, sleep deprivation. | Purification and protection. Public rituals and sacrifices. |
| Official *wu* (*siwu*, *nanwu*, *nüwu*)—Zhou dynasty (1046-256 BC) | Relate to royal ancestors, high gods (H, P) Knowledge of texts and rituals (H, P) Appease ancestors (P) | Not reported/ apparent (P, H) | Music, dancing, and drumming | Exorcisms (M, H) Sacrifices (M, H, P) Protection (P) Prayer (H) * |
| *Bo* of the *Tu* (Xing and Murray) | Protective, ancestor and tutelary spirits of village (P) Spirits not controlled-- may refuse to help (P, H, M) Procedures in hopes of coercing spirits Knowledge & skills | Possessed by spirits (M?) Become different spirits Spirits answer through mouth of *bo* (M) | Shivering with presence of a spirit (M?) Uses a drum and special chants and dances | Not source of help in illness (P) |
| Medium (M) (Etic Ritualist Type) | Possessing spirits dominate. Propitiation and sacrifices. Power dominates Unconscious. | ASC in training and practice. Possession ASC | ASC induced through singing, drumming, and dancing. Tremors, convulsions, seizures, compulsive behavior, amnesia, dissociation. | Possession and exorcisms. Control of possessing spirits. |
| Female Wu- Warring States period | Appealed to deities with sacrifices & prayers (P, M) Decision maker the possessing male spirit (M) Spirit or god descends on person (M) | Gods merged with personality of *wu* (M) Possessing male spirit or god descends on person (M) | Induction with dancing, incantations, singing and wailing | Spirit responsible for disease Perform exorcisms (M, H) Herbal healing Sacrifices, prayers, spells, incantations Anointing and ablutions |
| Sorcerer/ Witch (SW) (Winkelman Etic Ritualist Type) | Power from spirits and ritual knowledge. Contagious, exuvial and imitative magic, spells. Power can be unconscious, out of control. | Indirect evidence of ASC in reported flight and animal. transformation. | Night-time activities. | Illness by consuming victim's soul, spirit Aggression, magical darts that enter victim Unconscious emotional effects of envy, anger, etc. |
| *Wugu* | Incantations and Poisons Imitative magic (S, S/W) Cursing (SW) Burial of puppets | | Nighttime activities (S, SW) | Ambition, greed (S/W) |

Letters indicate contrastive features of practitioner type: S = Shaman feature, AS = Agricultural Shaman feature, M = Medium feature, H = Healer feature, P = Priest feature, SW = Sorcerer/Witch feature; ? = questioned by authorities; * = often attributed to ritualist type but not part of etic model.

Comparing Bureaucratic Wu and Priests (SCCS Variable 884)

The features of the bureaucratic *wu* and Priests are presented in Table 5. There are direct correspondences of the features of the Priests discovered in Winkelman's cross-cultural research with the characteristics of these *wu* state functionaries and bureaucratic ritualists central to Chinese religion and the clan institutions of ancestor worship. The Priests hold dual political–religious roles, with supreme secular and religious power generally invested in the group's highest-level Priest. The central societal roles of Priests give them high social status. The principal activities of Priests involve collective agriculture (or pastoral) rituals that propitiate group deities with public sacrifices and feasts. This worship of the gods is to secure well-being in agricultural or other economic activity through the performance of the most significant calendrical rituals at the crucial annual planting and harvesting cycles. The rituals associated with the agricultural cycle are intended to aid the fertility of animals and crops, and the harvest rituals focus on offering thanks to the deities for the abundance provided.

The supreme leader of the Priests is typically selected by patrilineal succession, although political action and armed conflict with competing relatives may be necessary to secure the top position. Priests are normally limited to males, with females generally serving only as assistants or servants unless it is the empress or queen who assumes the position of high Priest by virtue of her royal position or widowhood. The head Priest is assisted by diverse Priests who may have specialized training, often of a technical nature. ASCs typically were not considered significant in the training of Priests, but the training did involve periods of social isolation, sexual abstinence and prolonged wakefulness.

Priests are typified by ancestor cults led by the senior lineage member who assumes the role of the Priest for the group. Lower priests are senior lineage members and typically have power over extensive economic resources and often hold important political positions and exercise judiciary functions. Priesthoods are hierarchically organized in a system of administrative control over society. Priests may exercise legislative functions establishing the moral order of society. Priests obtain significant economic resources from their functions and their full-time profession.

Priests' power comes from their close ancestral relations with spirits of collective importance, especially ancestor spirits, village spirits and high gods. Priests are ritual intermediaries with the spiritual world, petitioning deities on behalf of the community, but do not normally enter into the spirit realms nor claim to control the gods. Priests' high prestige and personal power related to their divine descent may also give impersonal power such as mana (spiritual life-force energy). The Priest's ritual knowledge guides the timing of official public ceremonies and typically includes the sacrifice of domestic animals (cows, pigs, chickens, etc.) to the gods, which are consumed by participants in community-wide festivals. The collective rituals often involved collective alcohol consumption and feasting in prominent public places in collective ceremonies.

Healing is not a focal activity of Priests except in so far as to secure protection through purification and well-being from the worship of the gods through sacrifices. Although individual healing services were not Priestly functions, healing effects were provided through the psychological effects of purification and removing contaminated conditions such as offending spirits or taboo violations. Propitiating and worshiping the gods was also focused on well-being in seeking protection for the whole group, assuring abundance and preventing future illness.

The close correspondences of these bureaucratic functionaries and the Priests described in Winkelman's cross-cultural research illustrate furthermore that they do not remotely resemble Foraging Shamans. Michael asserts that the bureaucratic ritualists have "direct contact or face-to-face communication between humans and non-bodily beings in séance events" (Michael 2015, p. 671), suggesting ASC. This interpretation of the direct communication of these *wu* with spirits is questioned by Keightley, who notes that the spirits silently enter (descend) and leave the ritual after partaking in the sacrifices they have been offered.

"There is nothing particularly shamanistic about the quality of the religious experience described" (Keightley 1998, p. 10).

The Priests and bureaucratic *wu* share virtually every characteristic:

- primary religious activities (ancestor worship, agricultural well-being);
- selection (for the highest level, social succession/inheritance and political action);
- socio-political power (highest political, economic and judicial power);
- professional characteristics (male, full-time, specialized hierarchy, high status);
- motive and context of ritual (social functions, public calendrical rituals);
- supernatural power (ancestors, superior gods, ritual knowledge);
- ASC (very limited); and
- Healing (sacrifices).

On the other hand, the bureaucratic *wu* and Shamans differ on virtually all features besides high status. There is no empirical or rational justification to refer to the bureaucratic *wu* as Shamans. As Tong (2002, pp. 53, 60) notes "From then on, it is no longer proper to use the term *wu* to cover all the religious practitioners indiscriminately. Henceforth, part-time folk magicians may still be referred to as *wu*, but the religious elite of the community had become something closer to priests . . . . In their social status, activities, functions, and beliefs, they were in fact priests."

*3.5. Female Wu*

Literary sources from the Shang period emphasized the ancient distinctions of female *wu* from male practitioners, but there was little description of their activities in comparison to the male *wu xi* and *wu yi*. They were represented with a variety of characters (i.e., 巫尪 *Wu Wang* and 巫嫗 *Wu Yu*; also see Table 2). But by the fifth to third centuries BCE, only accounts of *wu* who were females were well documented in literary sources and government records because they held positions in the royal courts as advisors to supreme rulers (Sukhu 2012) and performed rituals for the state religion, including funeral rites (Michael 2015). The female *wu* were members of a coven who served the royal court through the ministry of ritual and accompanied rituals in the temples where they appealed to the ancestors and gods through sacrifices.

These *wu* were appointed to positions in the royal courts as advisors to supreme rulers and engaged in rituals involving dances and spirit communication to assure the well-being of the king and state, especially through rituals for rain which were seen as important for agricultural abundance. Cai (2014, p. 145) characterizes the various functions of these *wu* based on their abilities to communicate with the gods and other spirits whom they queried with divinatory rituals and petitioned with sacrifices and spells to obtain their favor. Female *wu* played important roles in political decisions with divination practices based on dream interpretation. But despite their employment by the king during the Qin and Han dynasties to make sacrifices to numerous spirits, especially ancestors, the social status of the female *wu* was generally low, a petty position comparable to court musicians and entertainers (Cai 2014, p. 145).

The ASC of female *wu* involved possession, a spirit communication in which a spirit descends from heaven upon the person who is possessed, with the personality of the *wu* entirely merged with that of the possessing male spirit. These experiences were also vital for receiving divinatory revelations by the *wu* which were important in political decisions, particularly decisions regarding war. Sukhu emphasizes their healing role as well, with interconnected skills of communicating with spirits for divination to determine the spirit responsible for a disease and treatments with exorcisms, sacrifices and herbal healing. Their healing rituals involved exorcisms carried out with incantations and sacrifices for appeasing the ancestors.

Comparing Female Wu with Mediums (SCCS Variable 882) and Shamans
(SCCS Variable 879)

The principal functions of Mediums involving healing, divination and spirit communication are the same as Shamans, but their features are generally different. While Mediums are similar to Shamans in dramatic experiences of ASC and an engagement with spirits, their ASC are distinct in experiences of possession by a powerful spirit or god, rather than the soul flight, death-and-rebirth and animal transformation of Shamans. This selection through possession experience characteristic of Mediums presents psycho-biological factors involving dissociation and trauma responses, including convulsions, tremors and amnesia (Winkelman 2018). Mediums' initial episode of spontaneous possession typically occurs in late adolescence or early adulthood and is manifested with tremors, convulsions, seizures, compulsive behavior and post-ASC amnesia. This episode is considered an illness caused by an affliction of a possessing spirit, and the required treatment initiates their training as a Medium. This training involves the ritual induction of ASCs through singing, drumming and dancing during which the patient learns how to control the possessing spirits and vocalize their demands. Once they achieve this control they are cured and begin to serve in the capacity of a healer for the community. These features of possession are not typical of Shamans.

Mediums lack the hunting ritual and sorcery features of Shamans. Instead, Mediums are involved in agricultural rituals and the propitiation of spirits and deities, especially in rituals to worship and propitiate their possessing spirits and make sacrifices to them to assure personal and collective well-being. Mediums are generally of low social and economic status but have respect among women and beyond since they express the personas of dominant male spirits. Mediums are subordinate to Healers and Priests but may exert social influence in the community because they manifest powerful spirits and convey their messages. Mediums have supernatural power by virtue of their possession; their bodies and voices are taken over by powerful spirits who communicate divine orders for others to follow. Mediums are not thought to control these entities, but to express their will. The Mediums' primary ritual involves entry into a possession state to allow the gods to work their influence. Mediums also attempt to influence these superior spirits through sacrifices.

Mediums' diagnosis of illness and healing involves exorcism to remove the influences of possessing spirits and protection from malevolent spirits and witches. Mediums produce their healing first through control of their own possession episodes, as well as through the general effects of ritual and ASC in eliciting endogenous healing responses (i.e., relaxation response, dissociation, placebo effects, hypnotic responses). These possession experiences produce personal transformations by allowing women to assume roles and express socially prohibited emotions. The widespread manifestations of similar possession phenomena illustrate it is not to be understood in cultural particulars, but instead as a manifestation of an evolved mechanism. This adaptation involves the compartmentalization of consciousness to accommodate to accepting long-lasting relationships and situations that are oppressive or even abusive (Winkelman 2018).

The features of female *wu* that are also typical of Mediums (but not Shamans) include the following: agricultural rituals, their control by possessing spirits, lower social status, predominantly female, a professional group, propitiation and sacrifices and healing involving exorcism. Lacking in these female *wu* are the central features of Foraging Shamans, such as powers derived from animals and a belief in their transformation into an animal; death and rebirth experiences during formation and an ASC of soul flight or journey; and healing practices involving the recovery of soul loss and the extraction of illness-causing sorcery objects. Furthermore, in those cases where the spirits descended into the *wu* and spoke "through the mouths and bodies of the *wu*" (Michael 2015, p. 684), we see evidence of possession rather than the shaman's ecstatic or out-of-body experience. Instead of the Shaman's control over spirits, the spirits have control over the possessed Mediums. Mediums are not typically associated primarily with animal spirits, but are instead important

to the gods, reinforcing Mediums' reputation as exclusively moral agents in contrast to the Shamans, who have moral duality balancing a reputation for both healing and sorcery.

In spite of some literary mentions of the female *wu*s' flights to heaven (Williams 2020), characteristics typical of shamanism of the traditions of north Asia, this soul flight feature of Shamans' ASCs is notably lacking in female *wu*. Rather than supporting Michael's argument for shamanism, the presence of possession is a clear indication of the female *wu* being Mediums rather than Shamans. Michael reviews evidence in Chinese commentarial precedents for this interpretation of possession in the classic texts in the word *jiang zhi* which means 'to descend and arrive', with the spirit descending into the person to possess the *wu* (also see Lin (2009, pp. 397–99), who argues that the meaning was clearly possession). Their identity as Mediums is further reinforced by the distinguishing ecological and social features of societies typified by Shamans (foraging) versus Mediums (agricultural, with warfare and political integration) exemplified by these female *wu* in a state-level agricultural society with rampant warfare between states (Warring States Period). Other principal characteristics of these *wu* during the Warring States era (paraphrased from Sukhu 2012) that correspond to features of Mediums but not Shamans included being only women, their focus on the ancestors and gods, and their experience of possession in training and professional practice, which they addressed with exorcisms.

### 3.6. Wugu as Malevolent Wu

Both earlier independent and bureaucratic *wu* underwent a further change at the end of the Han dynasty, a "radical transformation, in which they were systematically identified with the popular religion of the masses and became the targets of the active suppression by the functionaries both of the Confucians and Daoists and, a bit later, the Buddhists" (Michael 2015, p. 673). While Lin (2009) considers the criticism and doubts of social leaders to have damaged the social image of the *wu*, it was the political interdictions and attacks that led to their loss of political and social status. Although complete repression and interdiction did not occur in the pre-Qin periods, it was well-developed by the Warring States Period. After the transition from the feudal states to a unified imperial empire, this tendency to forbid practices of the *wu* accelerated, leading to a dramatic reduction in their status by the Han Period. Schafer also notes these dramatic changes in the *wu*, astutely observing that "After the Chou dynasty, the female shaman . . . was forced into sub rosa channels for the practicing of her magic arts, analogously to the witch of medieval Europe" (Schafer 1951, p. 134). Once the imperial system was in force, the *wu* fell into the lower classes and were not able to recover their former glory (Lin).

Although *wu* were generally considered to be benevolent, performing white magic, some texts indicate they also might engage in witchcraft. While this may have been a confusion with practitioners referred to in some texts as *wugu*, some texts have attributed malevolent practices to *wu* as well, who were employed by the powerful to cast evil spells on the victim (Sukhu 2012, p. 75). In contrast to the generally benevolent *wu*, "*Wugu* was the art of directing malevolent spirits to harm people" (Cai 2014, p. 146). The *wugu* performed malevolent rituals with the use of poisons and the invocation of evil spirits for personal reasons or to assist others to obtain power and wealth or take revenge against enemies (Cai 2014, p. 146). Inscriptions of oracle bones indicate practices of manipulating various insects and poisonous snakes to produce *gu* poison and the performance of nighttime rituals involving the manipulation of wooden dolls used to represent the intended victims (Cai 2014, p. 146). Accounts of these practices of *wugu* attest to ritual incantations to cause an evil spirit to invade a victim to cause illness and even death. Such practices were prohibited, and people accused were publicly executed.

The importance of witchcraft and the execution of witches came to the forefront when the aged and infirmed Emperor *Wu* accused a royal family of using a (foreign) *wu* to perform a curse and manipulation of sorcery dolls to kill him. They were arrested and tried and the entire family was executed. The notion that these practices came from outside of the culture was indicated by the prosecutor Jiang who likewise hired *wu* from central Asia

to find the sorcery items and exorcise the ghosts, leading to the arrest of suspects who were tortured and put to death (Cai 2014, p. 145).

Wugu as a Sorcerer/Witch (SCCS Variable 883)

The features of the *wugu* correspond to the type of ritualist Winkelman labeled as Sorcerer/Witch. The Sorcerer/Witch is devalued: an immoral aspect of the supernatural. These ritualists generally deny the accusations of being a Sorcerer/Witch. They are thought to be exclusively evil, to violate the moral order and even engage in acts of incest and cannibalism. The Sorcerer/Witch is found in complex agricultural societies and is significantly predicted by political integration and warfare. Sorcerers/Witches are considered to be exclusively malevolent, even causing illness and death to their own kin. A Sorcerer/Witch is often tortured and may be publicly executed. They are denounced for the destruction of economic resources, especially agriculture and livestock. The Sorcerer/Witch may perform their evil for clients but are generally thought to act out of revenge and for their own personal benefit or being motivated by negative emotions (anger, jealousy, envy, greed). Because of these emotions, the effects of the Sorcerer/Witch may operate unconsciously. They may, however, have learned techniques or acquired their power directly from their parents. Sorcerers/Witches exercise control over spirits and use techniques generally involving curses and spells; contagious, exuvial or imitative rituals; and discharges of power or darts that enter the victim. The Sorcerer/Witch engages in nighttime rituals and is believed to be able to fly or transform into an animal.

The *wugu* do exhibit many of these features—devalued, immoral, being publicly executed, acting out of envy and greed and using curses, spells and imitative magic. And like the Priests and Healers who hold government positions, governmental officials were central in designating who is this antithesis of morality and accusing, judging and executing those whom they considered to be guilty. In Winkelman's (1992) research, Sorcerers/Witches appear to generally be innocent victims, or local shamanistic healing traditions, rather than evil ritualists primarily engaged in malevolence. These competing ritualists are persecuted in a conflict between local culture and hierarchical power, reflected in the significant correlations of the Sorcerer/Witch with political integration beyond the local community—such as a kingdom or empire. Winkelman's model of the formation of the Sorcerer/Witch is therefore consistent with the actual nature of the ambiguous *wu* practices, being used for good (healing, worship) as well as for evil (curses, spells). Chinese history in the Han period provides clear evidence of this malevolent activity by certain parties, as well as the persecution of the guilty and innocent alike by government authorities.

*3.7. The Chinese Reindeer-Evenki Shaman*

If there is and was a "Chinese" shaman within the contemporary era, it clearly would include the ritualist of the Chinese Reindeer-Evenki, who appropriately use the indigenous term *šaman* to refer to their practitioners. Heyne (1999) integrated his own and others' research on their characteristics and activities; this is paraphrased and summarized below.

The principal religious activity of the *šaman* was the treatment of illnesses, divination and predicting the future and the performance of various sacrifices to ward off misfortune. The *šaman* also served as guides for the departed to the land of the dead and led seasonal and social rituals (e.g., annual celebrations, weddings and memorials for deceased persons). While the *šaman* was expected to act in an altruistic manner, they were also regarded with feelings of fear for their ability to perform malevolent rituals. Rival shamans engaged in magical battles that might end in the death of one of them.

The call to be a *šaman* came in mysterious illnesses, dreams and visions received from the tutelary spirit of deceased ancestors, reflecting the inheritance of the ability to become a shaman. Once they fell victim to the call, the initiate often became depressed, distracted and unresponsive and suffered attacks of hysteria accompanied by convulsions and periods of insensibility. When the clan system was still intact, the neophyte received orientation from an experienced shaman of another clan, but now the shamanic initiation always

occurred as self-initiation, as a purely individual endeavor. The *šaman* sought long periods of solitude and fasting in the wilderness where the person fell into a state of ecstasy and lost consciousness. In this state, they received the call and teachings of the spirit powers that gave the *šaman* the strength to conduct a new life through conversing with the spirits. During their long meditation, the candidate was killed and cut into pieces by the spirits who consumed the flesh of the initiate and afterwards reassembled the bones and brought the initiate back to life. When the initiate returned to the community the person was already a shaman.

The *šaman* was a person of great significance for the community but was never the head of the clan or a political leader. The *šaman* could be male or female and their part-time service did not generate any material advantage. The *šaman* had greater mental abilities and courage than other clan members: a capable and inspired person. But despite their altruistic service, the *šaman* had an ambiguous moral status based on the possibility that he or she might abuse their power someday. Nonetheless, their rituals were performed for the benefit of the whole community when the normal life of a group member was disturbed, or the group as a whole was falling into disorder. The community's feelings of confidence obliged the shaman, who could not avoid this duty once the call was received. The shaman's task required an entirely altruistic behavior and did not generate any material advantage.

The supernatural power of the *šaman* was derived from relations with animals, especially the red deer, whose natural and supernatural qualities were transferred to the costumed *šamans* as they rapidly danced around the fire. The *šaman* were thought to be capable of changing themselves into animals, i.e., into the animal-shaped bearers of their souls, such as bear, elk or reindeer. The *šaman* was a master of the spirits who were subservient and pliable to the needs of their shaman master. The *šaman* had command of techniques of ecstasy. Singing and drumming were the most important techniques for inducing the *šaman* to fall into a trance. The beat of the drum carried the Evenki shaman to mount the magic elk or deer to travel to the otherworld. The soul of the *šaman* traveled into the spirit world, changing into their alter ego in an animal shape world while lying unconscious on the floor. *Šamans* might also use alcohol to facilitate their spirit journey and increase their powers over spirits.

The healing concepts of the *šaman* involved belief in the malevolent influences of spirits which could cause many forms of misfortune. The *šaman* had to overcome and chase out the spirits. Soul recovery was used to recover the lost soul of a patient on its way to the realm of the dead and bring it back safely. The shamanic ritual, particularly the singing and drumming, had a hypnotic influence on the patient. The *šaman* provided protection for all members of the group with spiritual safeguards erected to help prevent abnormal states of mind and misfortunes, illness, accidents and suicide. The *šaman* also knew how to apply medicinal herbs and plants.

Comparisons of *Šaman* with Foraging Shamans (SCCS Variable 879)

Virtually all the exclusive features of Foraging Shamans are exhibited by the *šaman*. The healing and divination, as well as malevolent acts, are shared, as is the selection based on illness, dreams and visions from spirits. The formative period alone in the wilderness, with fasting and visions leading to death-and-rebirth experiences are shared features; as are the high status, lack of economic gain, both male and female part-time functionaries and the altruistic action on behalf of the community they were obliged to serve. *Šamans* controlled the spirits and their powers came from animals into which they were believed to transform for journeys to the spirit world. These and other classic features of shamanism and Shamans such as healing through soul recovery, removal of malignant spirits and use of medicinal plants all illustrate features of the *šaman* which correspond to Foraging Shamans.

The fact that the Chinese Reindeer-Evenki were hunters and pastoralists who immigrated to China around 200 years ago and constitute an ethnic group outside of mainstream

Chinese culture is instructive. It is only outside of the core of Chinese culture, in the tribal groups at the margins and periphery, that the Shaman is found in recent history.

### 3.8. Is the Contemporary Chinese Tu Bo a Shaman?

The information that Xing and Murray (2018) present on the religious system of the Tu ethnic group of Qinghai Province in Northwest China focuses on the *bo* 博, whom they translate as shaman, but the actual descriptions they provide make it clear that it has nothing to do with Foraging Shamans or Agricultural Shamans. Xing and Murray (2018) note the ambiguous meaning of shaman and the problems of being "exported somewhat willy-nilly to other religious specialists around the world ... [but] nonetheless use the term shaman" (p. 7 of 22). "What remains ambiguous, however, are the features that define shamans as a group, that justify a special label, and that distinguish them from other religious specialists" (p. 6 of 22), but they arbitrarily characterize the shaman as simply involving principally diagnosing and healing illness (also weather control and contacting spirits).

But in discussing the Chinese ethnic Tu and their ritualist *bo*, whom they label a shaman, they note the contrary, "that among the Tu there has been a radically diminished involvement by shamans in the diagnosis and healing of illness" (p. 14 of 22), and instead document the importance of the *bo* in agricultural (rain-making) rituals. The only additional features they attributed to justify the label of shaman is that the *bo* are part-time specialists who let spirits enter the community and are possessed by spirits to find solutions to individual, familial and collective problems through rituals of public drumming and special chants and dances. All the core features of Foraging Shamans are notably lacking.

While Xing and Murray suggest that the *bo* "fits neatly into the category of shaman," there is nothing about the *bo* that resembles the characteristics of the Foraging and Agricultural Shamans, as seen in the characteristics of the *bo* discussed below (also see Table 5). The authors explicitly recognize the inappropriate application of the term shaman to the *bo* in stating "This public climatological function of shamanic performance is compatible with, but differs in ritual emphasis from, the cross-culturally more common role of the shaman as a healer in private diagnostic and therapeutic sessions." So why call the *bo* a shaman? The features of *bo* enumerated in Table 5 illustrate no resemblance to Shamans but rather closely resemble Priests.

Xing and Murray are paraphrased below to characterize the principal religious activity of the *bo*. The *bo* officiates a festival to help ensure good weather, making agriculture possible by sending rain and protecting against excessive rain. The *bo* also performs rituals for life cycle events, such as weddings and for female fertility and childbearing. The authors emphasize that the *bo* no longer provides a principal source of help in illness.

The *bo's* political power is illustrated in their role in providing solutions to collective, familial and individual problems, and their leadership in collective rituals. Only males may become a *bo*, but while not a full-time specialist, it is a hereditary position passed down from one generation to the next within the same family (patrilineal succession). *Bo* function principally in an annual festival coordinated with the planting season and perform a public village festival in the temple where the entire community participates in rituals to protect from natural disasters and to secure blessings for the community.

The rituals solicit the protective spirits *longwang* (dragon kings) and the *niangniang* (queen mothers), benevolent figures associated with water and rainfall, as well as tutelary and ancestor spirits for the care of the village and agrarian plots. The ancestors are worshiped with rituals at ancestral altar gravesites with gifts. Notably, while the *bo* is ascribed special knowledge and skills that bring the spirits to the community, he does not control these spirits who may even refuse to keep their part of the ritual bargain. The *bo* is ascribed an ASC, being possessed by spirits during ritual dancing in which he becomes the various spirits who provide information through the mouth of the *bo*. The ASC is induced by a drum and special chants and dances but is only notable from shivering which indicates the presence of a spirit. While *bo* provided healing as a principal activity in the past, they are no longer a principal source of help in illness.

The authors note that in the past, females typically functioned as *bo*. "Tradition has it that in the past, not only could females function as bo, but that the bo were typically females. That is now emphatically no longer the case. The contemporary absence of females among the *bo* may be the result of a transition that came as an adaptation to Han Daoist influence" (p. 12 of 22). This suggests that the mediumistic functions of the *bo* were usurped by men, who brought more of a priestly role to the position, reflecting "syncretic incorporation of Han Daoist practices into the inventory and repertoire of the *bo*." (p. 12 of 22). The role of *bo* as Priests is illustrated in the most important contemporary role of the *bo* involving performance in a public festival whose principal objective is the recruitment of spirits to help in controlling the weather for assuring the success of agriculture.

### 4. Results: A Summary of Shared Features and Divergences

The analyses above have provided direct comparisons of the *wu* ritualists with Shamans and other etic types of ritualists. This section provides a summary assessment of the similarities of *wu*[7] based on the characteristics they share with Shamans or the other ritualists they most closely resemble. The comparisons focus on differentiating features, those characteristics which distinguish Shamans (i.e., soul flight and soul loss), as opposed to general characteristics shared by many types of ritualists (i.e., spirit interactions or healing). Thus, the comparisons are based on the following 19 unique or diagnostic features of Shamans:

- Ritual: (two variables) Community-wide ritual held over-night
- Distinctive Functions: (two variables) Malevolence/Sorcery, warfare
- Selection & training: (four variables) Dreams and illness as signs of spirit's request, vision quest alone in the wilderness, death-and-rebirth experience, self/spirit taught
- Social Characteristics: (three variables) Charismatic leader, informal power, male and female
- ASC: (two variables) Unconscious period, out-of-body (soul flight),
- Animal powers: (three variables) Animals as a personal and supernatural power, experiences of animal transformation, hunting magic
- Healing: (three variables) Soul loss and recovery, sucking/object extraction, the expulsion of attacking spirits (vs exorcism of possession).

Ancient *wu* (focused on the men = *xi* or *wu yi*, rather than women *wu*) have at most 2 of the 19 contrastive distinctive features of Shaman (perhaps out-of-body [soul flight] and animal powers). On the other hand, the Ancient *wu* exhibit 11 distinctive features characteristic of Healers and/or Priests. These include Priest—ancestor worship and hereditary aristocrat status; Healer and Priest—agricultural and livestock rituals, members of the ruling class, attendance of religious affairs of state, only males, the performance of sacrifices and prayer to gods for blessings to avert misfortune; Healer—use of statues and icons, mechanical system of divination i.e., bone oracles and healing through exorcism.

Commoner *wu* of the late Zhou period had 9 of the 19 contrastive characteristics of Shaman (Foraging and Agricultural), including malevolence/sorcery, warfare, dreams and illness as signs of spirit's request, charismatic leader, informal power, both male and female, out-of-body experience (soul flight), animals as personal and supernatural power and hunting magic (as well as many other Shaman features which were not unique to Shamans, i.e., part-time, ASC, communicate with spirits, no formal group, etc.). On the other hand, the Commoner *wu* had only four contrastive features characteristic of Healers or Priests.

State religious officials did not have any features unique to Shamans, but instead had virtually all of their features diagnostic of both Priests and Healers (12) or just Priests (six) or Healers (two).

Female *wu* of the Warring States period had none of the unique contrastive features of Shamans. Instead, the female *wu* had six[8] variable areas exclusive to Mediums (group training, male spirit or god descends on person, possessing spirit, gods merged with the personality of *wu*, low social status, principally or exclusively female); three that Mediums share with Priests or Healers (control rains for agriculture, perform exorcisms and

appeal to deities with sacrifices and prayers) and three characteristics typical of Priests (assure well-being of state, rituals held at regular times throughout year and contributed to bureaucratic and political decisions with divination).

*Guwu* of the Han period had only four characteristics shared by Shamans and the Sorcerer/Witch type (nighttime rituals, males and females, imitative magic [AS] and cause illness and death) and seven which were uniquely characteristic of the Sorcerer/Witch type.

The Chinese Reindeer-Evenki *šaman* had 15 of the 19 contrastive features characteristic of Shamans (community-wide ritual, malevolence/sorcery, dreams and illness as signs of spirit's request, vision quest alone in the wilderness, death and rebirth experience, self/spirit taught, charismatic leader, informal power, both male and female, out-of-body (soul flight) experiences, animals as a personal and supernatural power, experiences of animal transformation, healing of soul loss and recovery and expulsion of attacking spirits (vs exorcism of possession).

The *bo* of the Tu ethnic group of Qinghai Province Northwest China only has one feature diagnostic of Shamans (community-wide ritual activity), but eight characteristic features of Priests, three shared by Priest and Healers and three possibly indicative of Mediums. I say possibly because the assertion of possession for the *bo*, while a characteristic of Mediums, does not have the features of Mediums cross-culturally (convulsions, amnesia, erratic behaviors, etc.; see Winkelman 2018).

These assessments indicate that the etic categories for the Chinese ritualists are as follows:

- Ancient *wu xi* or *wu yi:* Healer
- Commoner *wu*: Agricultural Shaman
- State religious officials, bureaucratic *wu:* Priest
- Female *wu* of the Warring States period: Medium
- *Guwu* of the Han period: Sorcerer/Witch
- Chinese Reindeer-Evenki *šaman:* Foraging Shaman
- *Bo* of the Tu ethnic group of Qinghai Province: Priest

## 5. Discussion: Critical Assessment of Translating *Wu* as Shaman?

Determining whether a *wu* is a shaman or other type of ritualist should depend on their correspondence to cross-cultural patterns of shamanism and other religious ritualists described above (Tables 1, 3 and 5). Keightley (1998) made astute criticisms of the hypotheses that early Chinese ritualists were shamans in any way related to Eliade's conceptual framework. Williams (2020) further reviews studies (von Falkenhausen 1995; Keightley 1998; Boileau 2002) that have illustrated the error in labeling as shamans the various early Chinese religious practitioners referred to as *wu*. Yet many publications have promoted confusion by asserting a false equivalence of *wu* and shamans.

Michael (2015) notes "Confusion in using the category of shamanism arises in part because of a lack of consensus on which features to include in it, and that consensus can only be informed by adherence to culture-specific representations that then can be utilized in wider, cross-cultural studies" (p. 664). But it is the culturally specific allegations (definitions) of what is a shaman independent of some objective criterion that undermines efforts after consensus. We need cross-culturally valid concepts of shamanism such as those discovered in Winkelman's research, not some concept that loses comparative utility by changing from culture to culture.

But Michael seems to absolutely reject a cross-cultural definition of shamanism in asserting shamanism "does not exist as a natural piece of human behavior demonstrating essential qualities to be discovered and cataloged" (p. 665). The cross-cultural research of Winkelman and the biogenetic features shows that Michael is wrong. Cross-cultural studies can create a consensus about just what the concept of shaman represents, and whether it can be appropriately applied to culture-specific phenomena. A problem with Michael's approach is that he defines a shaman in a manner inconsistent with his own criteria. He alleges that "The presence of shamanism in any society is recognized by their representa-

tions of a séance event of direct contact (possession) or face-to-face communication (spirit journey) between human beings and bodiless beings for the benefit of the community" (p. 665). But such characterizations are so broad that it frustrates other aspects he emphasizes in his effort after a definition, which "also has to differentiate the shaman from all other religious and political functionaries; any definition that does not attend to this will inevitably be so open-ended that it will lose all tractability" (p. 659). The resolution of this problem of excessive open-endedness demands an ethnological perspective such as presented above, but which has not previously been applied to uses of the term *wu*.

Such cross-cultural characteristics of the empirically derived etic typology of ritualists determined by Winkelman's research provide an empirical framework to assess what is a shaman in a cross-cultural perspective and across Chinese history. The comparisons of this framework with Chinese ritualists presented above show that most types of Chinese ritualists called *wu* do not correspond to Foraging or Agricultural Shamans, but rather to other types of ritualists. This illustrates the necessity to distinguish the diverse types of ritualists called *wu* and relate them to cross-cultural patterns of ritualists besides shamans. Since the contrastive features of *wu* ritualists clearly map onto the ethnological cross-cultural model, it shows that such perspectives help articulate Chinese ritualists in comparative perspective and remove the erroneous perception that all *wu* are shamans. As illustrated above, while principal aspects of the features of the pre-historic and Commoner *wu* do correspond to the features of Foraging Shamans, the other types of *wu* (Ancient, State, Female and Wugu) do not correspond to the features of the Foraging & Agricultural Shamans, but rather other types of ritualists (Healers, Priests, Mediums and Sorcerers/Witches, respectively).

Boileau (2002, pp. 354–55) illustrates these diverse meanings of the *wu* character in its reference to many different types of ritualists and supernatural roles, including diviner, a possessed medium, a ritual scapegoat, a sorcerer, healer, priest and even ancestor worship practices and eventually activities suppressed by government officials who prosecuted and even killed *wu* during the imperial period for their practices, which were called sorcery and witchcraft. What these associations of *wu* with such diverse functionaries reflect is change, a socioeconomic transformation of ancient shamanic ritual practices into other kinds of ritualists.

Michael (2015, p. 685) proposes the experiences at the basis of Chinese state religion and centralized authority were "instituted, not by ancient, all-too human shamans or sages, but by the will of the spirits (speaking through the possessed *wu*) at a time before there were institutionalized priests and temple officers." This suggests that by the time of the early Chinese civilization with its state religion of ancestor worship, whatever may have been the early remnants of an aboriginal Chinese shamanism had already been transformed into Healers and Mediums and the Priests who controlled them. Since the Shang dynasty is considered the origins of Chinese civilization and was characterized by diverse forms of *wu* ritualists (male *wu yi*, female *wu* and official *wu*), Shamans were absent from the core of society at the beginning of Chinese civilization and only persisted at the peripheries of culture and power. This is exemplified in the two clearest cases of Chinese shamans presented here: (1) the Commoner *wu*, an Agricultural Shaman and (2) the Chinese Reindeer Evenki *šaman*, who reflects the classic features of Foraging Shamans.

Winkelman's (2022) cross-cultural analyses show that these processes involved agricultural intensification, supra-community political organization and eventually warfare. These processes lead to the elimination of Shamans and the assumption of religious and ritual functions by new types of ritualists that are not Shamans but rather Priests, Mediums and Healers. This was the dynamic present at the beginning of the Shang Dynasty—male *xi* Healers, the poorly documented female *wu* of that epoch and the bureaucratic *wu* Priests. This dynamic expanded in the Warring years with the development of the Female *wu* Mediums and the *Wugu* Sorcerer/Witch who was persecuted by the Chinese state. These dynamics of evolutionary change in Chinese ritualists are clearly reflected in the sociocultural evolutionary mode of religion presented in Figure 1.

What all the *wu* share is being a ritualist, not sharing the features of a Foraging or Agricultural Shaman. The use of the root *wu* in denominating diverse types of Chinese practitioners indicates that the best translation of *wu* is a religious ritualist.

*So What Is a Shaman?*

While cultural relativism might suggest that it makes sense to let each society define shamanism, there exists a problem that becomes clearer when asked in reference to other concepts. By analogy, should each culture or society decide what a democracy is? Does Putin's claim to run a democratic country receive obligatory acceptance by scholars of political systems? Or alternatively, should there be objective academic and scientific standards of what a democracy is?

By analogy, if we are to have cross-culturally relevant characterizations of the definition of a shaman, it must be based on empirical comparative data rather than arbitrary or culturally specific definitions. Determination of whether a ritualist is a shaman is not an issue of where the term originated or the culture where a particular ritualist is found, but whether the ritualist resembles a well-delineated cross-cultural phenomenon which justifies the etic use of a word as a transculturally relevant concept.

Concepts such as bands, tribes and chiefdoms are commonly used in comparative political assessments of the political complexity of societies. While each category shows variation, the differences among them are clear and useful. No informed anthropologist would confuse a band for a chiefdom. And even if people in chiefdoms band together for some reason, they are still a chiefdom, not a band. By analogy, the differences between Shamans and Priests are clear, not arbitrary. Winkelman's (1990, 1992, 2022) cross-cultural research found a consistent pattern of characteristics associated with the ritualists in foraging societies, and these features correspond closely to the core concepts identified by diverse scholars researching the nature of shamanism. This cross-cultural pattern of Foraging and Agricultural Shamans is the most objective criteria to use as a framework for characterizing and evaluating what a shaman was. And even if a priest heals, divines, talks to spirits, exorcises a patient and takes care of animals, his core features are that of a Priest, not a Shaman.

## 6. Conclusions: The *Wu* Is a Religious Ritualist, Not a Shaman

As illustrated in the ethnological studies presented above, there are consistent objective criteria about what the concept of shaman represents in foraging societies cross-culturally, and the consistent beliefs and behaviors associated with these ritualists are the criteria that should be used to determine whether the label shaman can be appropriately applied to culture-specific phenomena.

But despite decades of publications documenting the inappropriateness of the term shaman as a translation for *wu*, the use persists even among those who note that it is an inappropriate term. Fu's (2022) book has "shamanic" in the title, and in a chapter (6) on "The northern shaman", he writes about the well-preserved shamanic culture in myths and legends, and in spite of appearing to claim otherwise, clearly rejects the notion of any northern Chinese shamans: "on the basis of years of field surveys on the peoples of northern China, studies of shamans' biographies and historical documents, and interviews with the elderly and with old and new shamans, we can safely say that no northern Chinese shaman really performs practices that accord with Eliade's theories . . . While performing sacrificial rites for clan ancestors and nature deities, northern shamans remain conscious and sober-minded . . . . northern shamans do not actually lose consciousness, and their spirit certainly does not leave their body. Rather, they remain in control of the emotional vortex of the rite. In a large-scale ritual, their seeming trance state is actually a well-designed performance. In entering that state, their own human subjective activity, not divine power, is what is mainly at work" (Fu 2022, p. 158).

To appreciate the nature of Chinese *wu* ritualists in a cross-cultural perspective, particularly with respect to the concept of the shaman, we need what Feng Qu (2018) calls

for in assessing Chinese Mongolian ritualists, a two-way dialogue to resolve the problematic aspects of both Western and Chinese perspectives on what has been called shamanism. What is important is not just any Western perspective, but one informed by ethnology rather than the romantic traditions Qu criticizes. Clarity in academic discourse requires the unambiguous meaning of technical words. Translating *wu* as shaman obfuscates, confuses and misleads. A *wu* is a ritualist and may be a Shaman, but more likely is a Medium, Healer, Priest or even Sorcerer/Witch. Consequently, *wu* should be translated into English as "ritualist, or as "religious ritualist" if a distinction from other bureaucratic functionaries is needed.

**Funding:** This research received no external funding.

**Institutional Review Board Statement:** Not applicable.

**Informed Consent Statement:** Not applicable.

**Data Availability Statement:** Magico-religious-Practitioners data available at the Mendeley data repository at https://data.mendeley.com/datasets/34pjbr4kg4/2 Published 18 November 2020.

**Acknowledgments:** Many thanks to Liang Liu for assistance with Chinese characters, pinyin and English words used in Tables 2 and 4. Also thanks to the anonymous reviewers who requested revisions that contributed to the robustness of the methods employed and findings presented here.

**Conflicts of Interest:** The author declares no conflict of interest.

## Notes

[1]    For updated variables, values, variable descriptions, coding instructions and data see Winkelman and White (1987) or the Mendeley data repository at https://data.mendeley.com/datasets/34pjbr4kg4/2.

[2]    This access was made available by Doug White (RIP) but no longer appears available to the public.

[3]    Also referred to here as a Foraging Shaman.

[4]    Also referred to here as an Agricultural Shaman.

[5]    Winkelman (1992, p. 29) reports that the variables were attributed to a type if it was reported for 67% of the practitioners of the type or if the incidence of the variable for the practitioner type was at least 50% of all cases reported for the variable. In the case of Shamans, all cases of variables of less than 100% were correlated with data quality sources, with the consistently positive correlations indicating missing data (under reporting) rather than true absence.

[6]    Analyses were performed in the CosSci program housed at the University of California, Irvine at http://socscicompute.ss.uci.edu/. This system is no longer publicly available.

[7]    I have not provided a comparison with the Pre-historic Neolithic *wu* because of the lack of adequate data for a meaningful assessment.

[8]    Table 5 shows more than six matches with Mediums, but they are overlapping features.

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
