# Peer review of "Chinese Wu, Ritualists and Shamans: An Ethnological Analysis"

_religions, doi:10.3390/rel14070852_

Round 1
Reviewer 1 Report
the name Boileau is mis orthographied in two places.
the conclusion that wu should be translated by ritualist does not take in account the specifics of the wu since a lot of other functions (such as the shi historiographer and many other officiers described in the Zhouli and even bronze inscriptions did have ritual functions
Author Response
the name Boileau is mis orthographied in two places. Corrected
the conclusion that wu should be translated by ritualist does not take in account the specifics of the wu since a lot of other functions (such as the shi historiographer and many other officers described in the Zhouli and even bronze inscriptions did have ritual functions
This specification of wu as religious ritualists is added to the abstract paper and conclusions to make this clearer
Thank you for your suggestions.
Reviewer 2 Report
The article has serious flaws, additional experiments are needed, and research is not conducted correctly.
This article should be reconstructed. The flow of the arguments lacks coherence, and the organisational structure is poor, leading the readers hard to understand what the main points should be argued about and the presented research finding.
The article is long, but the content is presented in various subtopics and with lots of point forms. The author concludes: "Wu is a ritualist but not a Shaman", but it hardly to find the author's arguments with empirical evidence in the main text. I see in the article that most of the text evidence is based on other references. What is the originality of the author's work in the article? How do the author's opinions engage with the scholarship? What research finding is something new and valuable to be presented on this topic? The discussions are based more on other scholars' references and even listed out several of their points, making the article more of a literature review than a research work.
I suggest the author reconstruct the article structure into:
1. Introduction
2. Research Methodology
3. Research Results/Discussions
4. Conclusions
The language seems long-winded and repeatable should make it simplicate.
Chinese proper nouns in the article such as Wu(巫)suggested putting Chinese Characters.
Author Response
"Please see the attachment.

Reviewer 3 Report
The ambitions of this article are laudable and the effort to differentiate and nuance the category of wu in relation to its translation as ‘shaman’ important. However the argument of the paper is hard to follow: several lines of argumentation can be identified throughout the article, with unclear relationship among them. At this point the article reads as an elaborate draft where the direction of the main argument has not yet been decided upon, allowing for the disturbing intrusion of secondary arguments to occur.
On the one hand, we can see a line of thought which distinguished the historical transformation of the wu as related to his functions and role in society while on the other hand we see an effort to distinguish between the different types of wu as related to their clients position in society, distinctions which are a-historical, which implies that there is an inherent tension in the way these lines of thought are mixed in the article. These categorizations seem to be made on the basis of ethnographic literature, thus based on literature reviews, however, in the subcategories identified, the weight and comparison between sources is unequal (female wu subcategory relies exclusively on Cai’s account) and the relationship between mentioned sources is not always clear. Also not clear if all authors mentioned in a subcategory (for example bureaucratic wu priests) use this term themselves, or are the categories interpretations of the author of what the mentioned authors say. Thus it is not clear methodologically what is the procedure the author is using in this part and how much are the proposed categories interpretation.
On the other hand, next to the descriptive lines of argumentation we have several attempts at comparison, for example the comparison with Winkelman’s types of religious practitioners. This comparison lacks clarity: what is the comparison based on and how it is methodologically put into practice is not clear or consequent, some subsections introduce extensively new literature – the question remains, what is compared with what and what is the purpose of that comparison? Also, historicity and contextuality is completely missing from the comparative part, which makes the arguments confusing.
A thorough reworking of the structure of the article, with a clear methodological overview and a thorough explanation of the research question(s), an extensive literature review which allows the background but also contribution of this article to the light would be welcome. The extensive lists of attributes which occur often in the article disturbs the narrative of the article, a suggestion would be to opt for cursive text. Finally, a thorough language editing is suggested.
See above.
Author Response
"Please see the attachment.

Reviewer 4 Report
Review of “Which Wu Ritualist was the last Chinese Shaman?”
Caveats for this review: The version of the paper I have does not include a section on the biogenetic bases of shamanism that should start around line 794. Also Figure 1 and the table (?) starting on line 448 are jumbled and incomprehensible in the version I have. I do not think that these issues seriously impact my understanding of the authors’ arguments.
In this manuscript, the authors evaluate the appropriateness of using the term “shaman” as a translation for the term wu in the context of the study of Chinese magico-religious practices. The authors start the discussion by focusing on the development and structure of wu ritualist practices. Underlying this discussion is the application of a series of ritual categories developed by Mike Winkelman to classify ritualists into categories created using statistical analysis based on a sample of ethnographically studied cultures. The authors’ fundamental argument is that wu practitioners do not fit Winkelman’s derived category of “foraging shaman” and therefore ought to simply refer to generically as ritualists (see the final sentence of the manuscript—line 1124).
The majority of the manuscript outlines a chronologically ordered sequence of wu practices. It starts by noting that Eliade (1964) and others explicitly caution against considering wu as shamanism, but that in practice the term is regularly translated as “shaman.” Further, many scholars simply refer to nearly all Chinese ritualists as “shamans,” which the authors argue limits the utility of the term “shaman” while also mischaracterizing the Chinese ritual practices and practitioners. The authors then turn to summarizing Winkelman’s categories of religious practitioners based on his analysis of 47 societies contained in the Standard Cross-Cultural Sample. Winkelman defined 5 types of practitioners relevant for the authors’ evaluation here using cluster analysis: Foraging shaman, Agricultural shaman (Shaman/Healer), Priest, Healer, Medium, and Sorcerer/Witch.
The authors then describe and evaluate types of wu ritualists organized roughly chronologically.
· Commoner wu (earliest form of documented wu—provided healing, prayers, blessings, invocations against disaster, but potentially dangerous. Could communicate with gods and other spirits for divination. Exorcisms.
· Bureaucratic wu—Primarily responsible for needs of royal family, use of rituals dictated in written texts and performed according to a ritual calendar and in temples, full-time practitioners, relied on study and skills as opposed to supernatural gifts, high prestige, administered sacrifices, divination (using dreams), and exorcism. Sometimes paired with mediums.
· Female wu—Experienced possession and performed rituals as part of state religions. Some healing based on determining the spirit responsible for a disease.
· Wugu and Gadu—Low status. Persecuted by Confucians, Daoists, and eventually Buddhists. Wu could be practiced for good, but Wugu was the art if using wu for malevolent purposes. Used poisons and evil invocations. Could use dolls too. (Note: gadu is mentioned only once in the paper and has an unclear relationship to wugu or other religious practioners.)
The authors then compare each of these practitioners to the categories Winkelman defined. The authors determine that Commoner wu do not correspond perfectly with Winkelman’s shamanic characteristics: unclear if animals are involved, done for economic benefit, in a system that includes priests, no evidence of soul recovery. The authors suggest there is a closer match with agricultural shamans. Next, xi wu or archaic xi is discussed. Here the comparison shifts from comparing this wu practice with Winkelman’s categories to a more detailed comparison with Siberian shamans. Presumably the Siberian shamans are being used as an example of Winkelman’s “foraging shamans”, but it is unclear why the comparison is made to this specific tradition, as opposed to Winkelman’s more general discussion as is the case elsewhere. The shift in the basis of comparison is jarring. Regardless, the authors suggest the archaic xi reflect healers. Still, the correspondence between the archaic xi and “healers” is not perfect (see line 482 and thereabouts.)
The authors then propose there is a close correspondence between the bureaucratic wu and Winkelman’s category of “priest”. The next evaluations compare Wu (presumably female wu) and Wugu to Winkelman’s categories of mediums and sorcerer/witch. The authors suggest they fit within these categories although differences also exist. For example, the authors note that sorcerers/witches can “fly or transform into an animal,” but this is not demonstrated for wugu. Instead, what is discussed in lines 330 to 341 specific to the wugu are things like the use of poison, which are not stressed in the authors’ summary of Winkelman’s sorcerer/witch category.
The authors next provide a discussion phrased as a “Critical Assessment of Translating Wu as shaman.” The authors argue that wu is not a single ritualist system or category, but that the various forms of wu correspond to cross-cultural patterns. As stated in line 719 and thereabouts, “Since the contrastive features of wu ritualists clearly map onto the ethnological cross-cultural model, it shows that such perspectives help articulate Chinese ritualists in comparative perspective and remove the erroneous perception that wu are shamans.” The authors further suggest that shamanism can be defined through Winkelman’s cross-cultural approach. Although having talked about “agricultural shamans” elsewhere in the manuscript, this discussion limits the use of the term “shamans” exclusively to “foraging shamans.” The authors also include a critique of Michael’s (2015) definition of shamanism, suggesting it is too broad.
Next, the authors shift to evaluating if the contemporary bo tradition of the Tu people is shamanism. The authors do not explain the relationship between bo and wu. The authors note that Xing and Murray (2018) define shamanism as principally diagnosing and healing illness but also focused on weather control and divination/contacting spirits. The authors then criticize these authors for labeling the bo tradition as shamanism because it focuses primarily on weather control for agriculture, which seems to be logically inconsistent to me; I have not read Xing and Murray, but based on what this authors say, I am not sure I understand why Xing and Murray statement that shamans typically deal with healing and weather control is contradictory to their conclusion that in this case the bo prioritize weather control over healing. The authors of this manuscript then note that the bo tradition includes part-time specialists involved in spirit possession/spirit interaction but concludes the “there is nothing about the bo that resembles the characteristics of Foraging and Agricultural Shamans.” (lines 827 and 828) This does not seem to me to be correct, in that there are several features that seem to be shared with agricultural shamans as they are discussed elsewhere in the manuscript. However, there is no clear statement of the characteristics of agricultural shamans so it is hard to evaluate this claim. The authors then conclude that the bo practitioners ought to be considered priests, although I am not sure how this corresponds to the presence of spirit possession and other traits that I thought were not part of the Winkelman’s priest categories.
The authors then turn to the question of whether there is any evidence that wu practitioners ever corresponded to Winkelman’s foraging shamans and concludes that the answer is “no” in terms of the start of Chinese civilization, but that Chinese linguistic characters that perhaps reflect dancing shamans might indicate earlier foraging shamans.
Finally, the authors turn to the Chinese Reindeer-Evenki shaman that moved into China about 200 years ago. Their practices are distinct from wu but do correspond to Winkelman’s “foraging shamans” category. They ultimate conclude from this discussion that wu ought to be translated as “ritualist.”
My comments here are meant to be constructive, and I hope the authors take them in the spirit they are offered. I really like aspects of this paper and would like to see it published in an improved form. However, in its current form, I find this paper not yet ready for publication.
Here are some issues that struck me while reading the paper.
There is an inconsistency in the use of the term “shaman.” The authors frequently use the category of “agricultural shamans”, but then limit “shamans” to “foraging shaman” when doing so advances their argument. For example, Line 1090 states, “As illustrate in the ethnological studies presented above there are objective criterion about what the concept of shaman represented in foraging societies cross-culturally, and the consistent beliefs and behaviors associate with these cultures are the criteria that should be used to determine whether the label shaman can be appropriately applied to a culture-specific phenomena.” But what about “agricultural shamans” that are mentioned throughout the paper? These seem to be intentionally ignored as “shamans” in their analysis.
Winkelman’s categories of ritualists are not outlined with enough detail to evaluate his comparisons. Basically, the authors use a trait list approach based on Winkelman’s categories to show similarities with wu (and other) ritual practitioners. The traits for foraging shamans are presented in lines 358 to 383, but the traits for all other categories are never outlined with enough detail to allow the reader to really evaluate the argument. I realize the authors include citations to Winkelman’s original 1992 work and subsequent research, but that is not adequate for the reader to evaluate the arguments in this manuscript. The closest the authors get to providing a discussion of the different categories is in the section “Social Predictors of Ritualist Types” between lines 108 and 149. However, this discussion spends more time on economic/subsistence relationships than the actual defining characteristics of each category. If the author’s argument was based on subsistence and social organization (e.g., these practitioners must be agricultural shamans because they are in a society with intensive agriculture but without supra-community political integration), then this discussion would be relevant (if not particularly persuasive). However, the argument is instead based on the unspecified correspondence between list of traits associated with wu ritualists and those associated with Winkelman’s categories. That style of argumentation requires the reader to have detailed knowledge of the defining list of traits associated with each category, which simply is not present. Instead, differences between Winkelman’s categories of (foraging) shamanism and the traditions being discussed are held as hard and fast distinctions that demonstrate conclusively that the term/category of (foraging) shamans is not appropriate, even if there are apparent similarities. However, similarities with other terms/categories such as healers, priests, and mediums are held as conclusive proof that a practice fits into the categories even if there are apparent differences. This creates the impression of selective argumentation. This may not be the case, but without a clearer series of defining characteristics, it is impossible to evaluate. For example, this is clearly seen in the critique of Xing and Murray (2018) that comes across as arbitrary and in which clear similarities to the category of “shaman” and differences with the category of “priests” are both ignored to allow the authors to conclude bo practitioners are priests. This may be true, but the logic of the argument is underdeveloped.
The section “So What is a Shaman?” (lines 761 to 790) was unpersuasive to me. The authors' argumentation here seems to be that there must be universal features used to define shamanism given there are universal features used to define democracy, child abuse, and women’s rights. This logic seems flawed to me. In terms of the specific comparative examples, there is tremendous cross-cultural variation in terms such as “child abuse” for example with cultures ranging in cultural acceptance of childhood sexual practices, age at marriage, arranged marriage, disciplinary practices, genital modification, and so forth. However, even if one were to accept that “women’s rights” or “child abuse” had universal, clearly defined criteria, it would not be necessary to accept that the term “shaman” must also have universal, clearly defined criteria. Anthropologists readily acknowledge that we use terms like “chiefdom” that may reflect general characteristics but that are heuristic as opposed to essentialist in their definition. Why can’t this also be true for terms related to religious/ritual practices?
Finally, the manuscript’s focus shifts from section to section and is not consistent in theme or argumentation. I mentioned previously the shifting focus on comparison to Winkelman’s categories for commoner wu to Siberian shamanism for the archaic xi. Other examples include the additions of bo ritualism by the Tu and Chinese Reindeer-Evenki shaman that are distinct from wu. These discussions only make sense if the topic being evaluated is whether any Chinese traditions correspond to “foraging shamans” specifically, as opposed to the stated topic (in the title and the introduction) of whether wu practitioners are shamans.
Having provided what sounds like a harsh evaluation, please let me stress that there is a potentially interesting paper that could be developed from this manuscript with some reorganization and shift in focus. Forgive the tortured analogy but in some ways, this paper is like someone arguing that not all fruit are oranges, and then outlined different types of fruit found in a region and then concluding, “see, they aren’t all oranges.” This is a very narrow point that would only be interesting to someone who is exclusively focused on the study of oranges. A far more interesting analysis would be to focus on the actual range of fruit and their significance. Likewise, here the authors’ focus on comparing the traditions to (foraging) shamans limits the insights and appeal of the research. Their analyses are most persuasive and interesting when they focus on what the wu (and other) practitioners are, as opposed to what they are not. In other words, I learned the most from this paper when the focus was on everything else other than the authors’ stated focus of determining whether wu should be translated as “shaman.” If the authors need to maintain a focus on shamanism because this manuscript is part of a special themed issue or for some other reason, they could still accommodate a focus on shamanism by change their title/focus to something like, “Wu and its development from Foraging Shamanism”. They already seem to accept that foraging shamans were present in China before the development of Chinese civilization, so they could start with their linguistic evidence and go from there. That would sill leave the bo and Chinese Reindeer-Evenki discussions poorly integrated given they are not part of the wu traditions (at least as presented by the authors of this manuscript), but it would free the author from having to make negative arguments about what the wu traditions aren’t as opposed to furthering our knowledge of what they are.
Also, I understand that the authors find Winkelman’s categories useful, but their reliance on these categories needlessly complicates their discussion. I like Winkelman’s work, but I have always thought of his categories as more heuristic than deterministic in their utility. It seems unreasonable to me to suppose that Winkelman’s 115 culturally defined types of religious practitioners from only 47 historically known societies could reflect 100% of the variation present in past and even current cultures. It doesn’t surprise me that the wu ritualists apparently do not fit his categories perfectly. Could the authors shift their argumentation a bit to include Winkelman’s categories but work with them more loosely as opposed to trying to shoehorn everything into single categories? This approach would let them build from Winkelman’s framework to increase understanding of wu practitioners by allowing us to see how wu ritualists are similar to other traditions around the world while also allowing us to see how they might differ from what is cross-culturally common.
Author Response
"Please see the attachment.

Reviewer 5 Report
I am concerned with the use of "magic" or "magico-" in the paper - a term like magic does not give the religion the level of respect that the journal should expect. Ritual is much better and I note that magic is not used that often, so it should be dropped. Also - can the issue of s/he be resolved when females and males could perform the fuctions - perhaps s/he or they? He in some sentences and she in others does not work.

I attach a file with amendments that I suggest and a few questions. English is my first language.
Author Response
"Please see the attachment.

Round 2
Reviewer 2 Report
The author did a very detailed reconstruction of his/her article.
The article looks good after the amendment. By using a broader view to discuss Wu Ritualist.
I am satisfied with the author's work.
Author Response
thank you
Reviewer 3 Report
The article has greatly improved in coherence of argument.
Author Response
thank you I did not see any further specific requests
Reviewer 4 Report
This version is much improved compared to the previous version I read. You are asking for a quick response so I do not provide a page by page detailed review of the manuscript, but the authors' arguments are much clearer and the paper is better structured. Bottom line, the paper is publishable, but "average" in my opinion.
I find the authors' arguments plausible but not fully compelling. The reliance on Winkelman's arguments only really works if one already accepts the categories, but it seems to me that some aspects of the categories are circular to a degree. For example, foraging shamans and agricultural shamans are defined in part by differences inherent to foraging and agriculture subsistence (e.g., agricultural shamans focus on agricultural productivity). This is true, and I can accept these sorts of differences in a descriptive framework. However, it seems to me to be reasonable to say that foraging shamans, agricultural shamans, and healers are fundamentally similar, with the primary difference being that they are embedded in different subsistence/political systems. In other words, unlike the authors who seem to follow Winkelman in holding that foraging, agricultural shamans, and healers are different kinds of religious practitioners, it seems to me that the similarities reflected in Table 1.1 indicate they are really the same type of practitioners working in different subsistence/social systems. However, it is clear that this is simply an area where the authors and I disagree (to at least some extent), which is absolutely great. Differences in perspectives lead to academic growth.
So, putting aside our differences in perspective, I find the revised paper better organized and better argued than the original version. It is logically coherent, and the new tables help a lot. As a result, I find the revised version publishable in the sense that I understand the authors' arguments clearly and the authors' arguments are logically coherent and reasonable given the literature they cite. The manuscript works.
Author Response
Related to this statement, I think there is a minor understanding. "foraging shamans and agricultural shamans are defined in part by differences inherent to foraging and agriculture subsistence (e.g., agricultural shamans focus on agricultural productivity)."
Foraging shamans and Agricultural shamans are not "defined". Rather they were derived from empirical analyses. And the variable used were not based on anything inherent to foraging or agricultural societies. The differences were derived from the characteristics of their ritual activities, and relationships to foraging or agriculture were only assessed if they were part of the ritual activities. So there is not a circular or conflated argument here, rather the external relationships to agriculture and foraging are a form of validity for the independently derived groups based on empirical analyses of shared features or ritualists.
Certainly healers, mediums, f&A shamans share some similarities-- spirit relations, healing, etc. but their differences are considerable in spite of some shared functions.
But their similarities is why I have proposed shamanistic healers as a general term rather than shamans.
it seems that you are satisfied with my changes, I did not see any further items to respond to. Thanks for your feedback